J Physiol 603.4 (2025) pp 929–947

# Inhibition of diacylglycerol lipase α induced blood–brain barrier breach in female Sprague–Dawley rats

Erika Liktor-Busa[1], Aidan A. Levine[1], Sally J. Young[1], Colin Bader[1], Seph M. Palomino[1], Felipe D. Polk[2], Sarah A. Couture[1], Paulo W. Pires[2], Trent Anderson[1] and Tally M. Largent-Milnes[1]

[1]*College of Medicine, Department of Pharmacology, University of Arizona, Tucson, AZ, USA*
[2]*College of Medicine, Department of Physiology, University of Arizona, Tucson, AZ, USA*

Handling Editors: Harold Schultz & Nikki Jernigan

The peer review history is available in the Supporting Information section of this article (https://doi.org/10.1113/JP287680#support-information-section).

**Abstract figure legend** This study demonstrated that intraperitoneal administration of the diacylglycerol lipase α (DAGLα) inhibitor, LEI-106, causes blood–brain barrier (BBB) disruption manifesting as increased [$^{14}$C]sucrose uptake in *in situ* brain perfusion experiments. In the cerebral cortex, LEI-106 directly inhibited the production of the endocannabinoid 2-arachidonoylglycerol (2-AG) via DAGLα, causing the loss of 2-AG's protection on the BBB. The opening of the BBB in cortical areas was accompanied by decreased detection of the tight junction protein zonula occludens-1 (ZO-1). Optical measurements revealed increase in cerebral surface blood flow after DAGLα blockade. The administration of LEI-106 also resulted a significant elevation in pressure-induced contractility of penetrating arterioles, without affecting pial arteries. These data support the role of endogenous 2-AG in maintenance of tight junctions and blood–brain barrier function.

**Abstract** The endocannabinoid system's significance in maintaining blood–brain barrier (BBB) integrity under physiological and pathological conditions is suggested by several reports, but the underlying molecular mechanisms are not well understood. In this paper, we investigated the effects of depletion of 2-arachidonoylglycerol (2-AG), one of the main endocannabinoids in the central nervous system, on BBB integrity using pharmacological tools. Female Sprague–Dawley rats were injected with the diacylglycerol lipase $\alpha$ (DAGL$\alpha$) inhibitor LEI-106 (40 mg/kg, I.P.), followed by assessment of BBB integrity via *in situ* brain perfusion. Liquid chromatography–mass spectrometry, western immunoblotting, light transmittance experiments and pressure myography were also used to further examine the results of DAGL$\alpha$ blockade on the BBB and vascular reactivity. We found that DAGL$\alpha$ inhibition caused BBB opening in cortical brain areas, manifesting as increased sucrose transport measured by *in situ* brain perfusion. This was accompanied by reduced levels of 2-AG and decreased detection of the tight junction protein zonula occludens-1 (ZO-1). The protein level in cortical areas of neuronal PAS domain protein 4 (NPAS4), encoded by an activity-dependent immediate early gene, was increased without the presence of cortical spreading depression after LEI-106 administration. We also observed a significant increase in pressure-induced constriction within the parenchymal microcirculation after inhibition of DAGL$\alpha$, possibly altering shear stress in the microcirculation. These results support the role of endogenous 2-AG in maintaining normal tight junction function. This improved understanding of the molecular mechanisms of endocannabinoid system function at the neurovascular unit can help to unlock the therapeutic potentials of cannabinoids in central nervous system disorders associated with BBB dysfunction.

(Received 11 September 2024; accepted after revision 20 December 2024; first published online 18 January 2025)

**Corresponding author** T. M. Largent-Milnes: Department of Pharmacology, University of Arizona, 1501 N. Campbell Ave., Tucson, AZ 85719, USA. Email: tlargent@arizona.edu

## Key points

- The administration of the diacylglycerol lipase $\alpha$ (DAGL$\alpha$) inhibitor LEI-106 (40 mg/kg, I.P.) induced blood–brain barrier (BBB) opening of cortical brain areas in female Sprague–Dawley rats.
- This BBB disruption was accompanied by reduced levels of 2-arachidonoylglycerol (2-AG) and decreased detection of the tight junction protein zonula occludens-1 (ZO-1).
- The protein level in cortical areas of neuronal PAS domain protein 4 (NPAS4), encoded by an activity-dependent immediate early gene, was increased without the presence of cortical spreading depression after LEI-106 administration.
- A significant increase in pressure-induced constriction within the parenchymal microcirculation was also observed after inhibition of DAGL$\alpha$, possibly altering shear stress.
- These results support the role of endogenous 2-AG in maintaining normal tight junction function.

## Introduction

The blood–brain barrier (BBB) tightly regulates the influx and efflux of ions, molecules and cells between the blood and the brain, and is thus critical for the maintenance of central nervous system (CNS) homeostasis and function. This dynamic interface between the blood circulation and the neural tissue is a highly controlled, yet fragile microenvironment. Disrupting BBB integrity can cause or contribute to the pathology of numerous neurological diseases, including stroke, epilepsy, traumatic brain injury and headache (Profaci et al., 2020).

The endocannabinoid system (ECS) comprises two primary endocannabinoid receptors, CB1 and CB2, bioactive endocannabinoid ligands, 2-arachidonoylglycerol (2-AG) and *N*-arachidonoylethanolamine (AEA) and the enzymes responsible for their synthesis and metabolism (Lu & Mackie, 2021). Endocannabinoids are produced 'on demand' and are triggered by increased intracellular $Ca^{2+}$ at postsynaptic sites. The endocannabinoids synthesized postsynaptically exert their effect on cannabinoid receptors expressed on the presynaptic terminal in a signal mechanism called retrograde neurotransmission (Cristino et al., 2020)

The neuroprotective role of the ECS under different pathological conditions is widely accepted, and increasing evidence suggests that it plays a role in the preservation of BBB homeostasis (Hagan et al., 2022; Vendel & de Lange, 2014). A review published by Vendel and de Lange (2014) suggests that cannabinoid receptor-mediated protection of the BBB is partially exerted indirectly though reduction of excitotoxicity, cell death, inflammation and oxidative stress, all processes that can potentially damage the BBB. The direct effect of cannabinoid signals on the BBB has also been proposed, but the molecular mechanisms of this direct effect are not well understood. Activation of cannabinoid receptors has been shown to improve the integrity of the BBB by restoring the stability of tight junctions. Non-specific cannabinoid receptor agonists and selective CB2 agonists have mitigated the loss of the tight junction proteins zonula occludens-1 (ZO-1), junctional adhesion molecule-1 (JAM-1) and claudin-5 after inflammation in brain microvascular endothelial cells (Ramirez et al., 2012). Altered endocannabinoid metabolism, such as reduced plasma levels of 2-AG and AEA, accompanied with higher expression of their respective metabolizing enzymes, monoacylglycerol lipase (MAGL) and fatty acid amide hydrolase (FAAH), has been found in a rodent models of traumatic brain injury that are known to have compromised BBB integrity (Ahluwalia et al., 2023). Altogether, these results support the role of the ECS in the maintenance of BBB homeostasis. Nevertheless, how the ECS exerts this effect at the level of multicellular vascular structure remains undefined.

Previously, we showed that depletion of 2-AG by pharmacological or genetic inhibition of diacylglycerol lipase $\alpha$ (DAGL$\alpha$), the primary enzyme responsible for 2-AG synthesis from 1,2-diacylglycerol in the CNS (Reisenberg et al., 2012), caused breaches in the integrity of cultured brain endothelial cell monolayers (bEnD.3 cells), increased paracellular transport and caused fragmentation of the tight junction protein VE-cadherin (Levine et al., 2024). To further elucidate the effects of 2-AG depletion on the BBB, in this paper we tested whether DAGL$\alpha$ inhibition induced BBB breaches *in vivo*, as well as its impact on neuronal excitability and microvascular reactivity.

## Methods

### Ethical approval

The study is in accordance with the *Guide for the Care and use of Laboratory Animals* published by the US National Institutes of Health and it follows the recommendations of the International Association for the Study of Pain. All experimental procedures were approved by the Ethical Committee of the University of Arizona (protocols nos: 17-223 and 18-473) and comply with the policies of *The Journal of Physiology*.

### Drugs

Ketamine, xylazine and isoflurane were purchased from VetOne (Boise, ID, USA). LEI-106, 2-AG, 2-AG-d5 and AEA-d4 were purchased from Cayman Chemicals (Ann Arbor, MI, USA). Animals received LEI-106 (40 mg/kg, I.P.) dissolved in dimethyl sulfoxide (DMSO)–Tween 80–0.9% saline (1:1:8, v/v/v) 90 min prior to analysis.

### Animals

Intact female Sprague–Dawley rats (7–8 weeks old) were purchased from Envigo (Indianapolis, IN, USA) and housed in a climate-controlled room on a regular 12/12 h light/dark cycle with lights on at 07.00 h with food and water available *ad libitum*. Animals were housed three per cages. Animals were handled daily for a minimum of 5 min each and housed within the vivarium for at least 1 week prior to harvest. Daily vaginal smear and microscopic evaluation for the oestrous stage were performed as previously described (Cora et al., 2015), and those animals in the dioestrus phase were utilized for perfusion and harvest. It was ensured that animals had progressed normally through the oestrous stages (pro-oestrus, oestrus, metoestrus and dioestrus) for at least 8 days prior to perfusion and tissue harvest. Vaginal cytology was assessed at approximately 09.30 h each day, with perfusion and tissue harvest taking place at 11.00 h. Numbers required to achieve statistical power were determined by G.Power3.1.

### *In situ* brain perfusion

*In vivo* perfusion studies were carried out in intact female Sprague–Dawley rats (200–250 g) 90 min following LEI-106 (40 mg/kg, I.P.) administration. Briefly, rats were anaesthetized with ketamine–xylazine (80:10 mg/kg, I.P.) and heparinized (10,000 U/kg I.P.). The depth of anaesthesia was monitored by checking withdrawal and corneal reflexes along with monitoring the oxygenation and cardiorespiratory function (presence of pink mucous membranes, observing chest wall movement, rate and character of respiration) at the beginning and throughout the surgical process (every 5 min). Body temperature was maintained at 37°C using a heated pad. The common carotid arteries were cannulated with silicone tubing connected to a perfusion circuit then perfused with an erythrocyte-free modified mammalian Ringer solution (in mM: 117 NaCl, 4.7 KCl, 0.8 MgSO$_4$, 1.2 KH$_2$PO$_4$, 2.5 CaCl$_2$, 10 D-glucose, 3.9% (w/v) dextran ($M_r$ 60,000) and 1.0 g/l bovine serum albumin (type IV), pH 7.4), warmed

to 37°C and oxygenated with 95% $O_2$–5% $CO_2$. Evans blue dye (55 mg/l) was added to the perfusate to serve as a visual marker of BBB integrity. Perfusion pressure and flow rate were maintained at 95–105 mmHg and 3.1105 105 ml/min respectively. Both jugular veins were severed to allow for drainage of the perfusate. Using a slow-drive syringe pump (0.5 ml/min per hemisphere; Harvard Apparatus, Holliston, MA, USA), [$^{14}$C]sucrose (0.5 µCi/ml) was added to the inflowing perfusate. Following a 15-min perfusion, the rats were decapitated, brains were removed and the cortices, periaqueductal grey (PAG) and medulla were isolated. The meninges and choroid plexus were excised, and cerebral hemispheres were sectioned and homogenized. TS2 tissue solubilizer (1 ml) was added to each tissue sample and the samples were allowed to solubilize for 2 days at room temperature. To eliminate chemiluminescence, 100 µl of 30% glacial acetic acid was added, along with 2 ml Optiphase Super-Mix liquid scintillation cocktail (PerkinElmer, Waltham, MA, USA). Samples were measured for radioactivity on a model 1450 liquid scintillation counter (PerkinElmer).

## Tissue harvest

Rats were anaesthetized with ketamine–xylazine mix (80:10 mg/kg, I.P.), then transcardially perfused with ice-cold 0.1 M phosphate buffer at flow rates to not burst microvasculature (i.e. 3.1 ml/min). The depth of anaesthesia was monitored by checking withdrawal and corneal reflexes along with monitoring the oxygenation and cardiorespiratory function (presence of pink mucous membranes, observing chest wall movement, rate and character of respiration) at the beginning and throughout the process. After decapitation, PAG, cortex, medulla and blood samples were harvested. Tissue samples were flash-frozen in liquid nitrogen and stored at −80°C until further use. Blood samples were centrifuged at 3800 $g$ for 15 min at 4°C, then supernatant as the serum was collected and stored at −80°C until further use.

## Tissue preparation for western immunoblotting

On the day of preparation, samples were placed in ice-cold lysis buffer (20 mm Tris–HCl, 50 mM NaCl, 2 mM $MgCl_2.6H_2O$, 1% v/v NP40, 0.5% v/v sodium deoxycholate, 0.1% v/v SDS; pH 7.4) supplemented with protease and phosphatase inhibitor cocktail (Halt$^{TM}$ Protease and Phosphatase Inhibitor Cocktail, Thermo Fisher Scientific, Waltham, MA, USA). All subsequent steps were performed on ice or at 4°C. The samples were sonicated and then centrifuged at 12,000 $g$ for 10 min. The supernatant was collected from the samples and a BCA assay was performed to determine the protein content (Pierce BCA Protein Assay Kit, Thermo Fisher Scientific).

## Western immunoblotting

Twenty-five micrograms of total protein was loaded into TGX precast gels (4–20% Criterion, Bio-Rad Laboratories, Hercules, CA, USA) and transferred to nitrocellulose membrane (Amersham ProtranTM, GE Healthcare, Chicago, IL, USA). After transfer, the membrane was blocked at room temperature for 30 min in a blocking buffer (5% dry milk in Tris-buffered saline (TBS)). The following primary antibodies were used: neuronal PAS domain protein 4 (NPAS4) (Thermo Fisher Scientific, MA5-27592, 1:500), VE-cadherin (Thermo Fisher Scientific, 36-1900, 1:300), claudin-5 (Thermo Fisher Scientific, 35-2500, 1:300), ZO-1 (Thermo Fisher Scientific, 40-2200, 1:300), cFOS (Abcam, Waltham, MA, USA, ab190289, 1:1000), $\beta$-actin (Abcam, ab8226, 1:2000) and $\alpha$-tubulin (Cell Signaling Technology, Danvers, MA, USA, 3873S, 1:10,000). The primary antibodies were diluted in 5% BSA in TBST (TBS with Tween 20). The membrane was incubated in diluted primary antibodies for 48 h at 4°C. The membrane was then washed three times in TBST for 5 min each followed by incubation with IRDye 800CW Donkey anti-Rabbit IgG Secondary Antibody (Li-Cor, Lincoln, NW, USA, 926-32213) and IRDye 680RD Donkey anti-Mouse IgG Secondary Antibody (Li-Cor, 926-68072) in 5% milk in TBST for 1 h rocking at room temperature. The membrane was washed again two times for 5 min each in TBST. TBS buffer was used for the last washing step. The membrane was imaged with an Azure Sapphire laser imager (Azure Biosystems, Dublin, CA, USA). Un-Scan-It 6.1 software (Silk Scientific Inc., Vineyard, UT, USA) was used for quantification.

## Liquid chromatography–mass spectrometry

Tissue samples for liquid chromatography–mass spectrometry (LC-MS) were purified by organic solvent extraction with a protocol modified from Wilkerson et al. (2016). Briefly, pre-weighed samples were homogenized in 1 ml of chloroform/methanol (2:1 v/v) supplemented with phenylmethylsulfonyl fluoride (PMSF) at 1 mM final concentration to inhibit the degradation by endogenous enzymes. Homogenates were then mixed with 0.3 ml of 0.7% w/v NaCl, vortexed and then centrifuged for 10 min at 3200 $g$ at 4°C. The aqueous phase plus debris was collected and extracted two more times with 0.8 ml of chloroform. The organic phases from the three extractions were pooled and internal standard was added to each sample. Mixed internal standard solutions were prepared by serial dilution of AEA-d4 and 2-AG-d5 in 80% acetonitrile. The organic solvents were evaporated under nitrogen gas. Six microlitres of 30% glycerol in methanol per sample was added before evaporation. Dried samples were reconstituted with 0.2 ml of chloroform and mixed with 1 ml of ice-cold acetone to precipitate proteins. The

mixtures were then centrifuged for 5 min at 1800 *g* at 4°C. The organic layer of each sample was collected and evaporated under nitrogen.

After measuring the volume of serum samples, they were mixed with 1 ml of chloroform–methanol (2:1 v/v) supplemented with PMSF at 1 mM final concentration. Samples were then centrifuged for 10 min at 3200 *g* at 4°C. The aqueous phase plus debris was collected and extracted two more times with 0.8 ml of chloroform. Subsequent steps were performed as described above.

Analysis of 2-AG and AEA was performed on an Ultivo triple quadrupole mass spectrometer combined with a 1290 Infinity II UPLC system (Agilent Technologies, Santa Clara, CA, USA). A detailed description of the experimental setting was published in our previous paper (Levine et al., 2021).

## Quantitative real-time PCR

The total RNA of cortex samples was extracted using RNeasy Plus Mini Kit (Qiagen, Germantown MD, USA, 74134) per the manufacturer's protocol, with final reconstitution of the RNA in nuclease-free water and storage at $-80$°C until further use. Single-stranded complementary DNA (cDNA) was synthesized using a high-capacity cDNA reverse transcription kit (Thermo Fisher Scientific, 4368814), following the manufacturer's protocol. The cDNA was quantified using a NanoDrop ND-1000 Spectrophotometer (Thermo Fisher Scientific). The amplified cDNA was added to the $RT^2$ SYBR Green Master Mix (Qiagen, 330500) and gene-specific primers. The following $RT^2$ qPCR primer assays were applied: $RT^2$ qPCR Primer Assay for Rat Fos (Qiagen, PPR55248C-200), $RT^2$ qPCR Primer Assay for Rat Npas4 (Qiagen, PPR52619A-200) and $RT^2$ qPCR Primer Assay for Rat Gapdh (Qiagen, PPR06557B-200). The quantitative PCR (qPCR) was performed on the StepOnePlus Real-Time PCR System (Thermo Fisher Scientific), using the thermal program: 95°C for 10 min, 40 cycles of 95°C for 15 s and 60°C for 1 min. Cycle thresholds ($C_t$) were normalized to the $C_t$ value of *Gapdh* to generate relative gene expression of each target.

## Light transmittance experiments: slicing procedure

Adult female Sprague–Dawley rats (age 50–80 days) were anaesthetized with 5% isoflurane and killed by decapitation. The brains of the rats were harvested in a $\sim$4°C slicing solution (in mM: 234 sucrose, 11 glucose, 26 $NaHCO_3$, 2.5 KCl, 1.25 $NaH_2PO_4.H_2O$, 10 $MgSO_4.7H_2O$ and 0.5 $CaCl_2.2H_2O$), and 350 µm slices were taken from the region containing the visual cortex (V1) (Paxinos & Watson, 2013). The slices were bisected and incubated in carboxygenated (95% oxygen–5% carbon dioxide) artificial cerebral spinal fluid (aCSF) (in mM: 126 NaCl, 26 $NaHCO_3$, 2.5 KCl, 10 glucose, 1.25 $Na_2H_2PO_4.H_2O$, 1 $MgSO_4.7H_2O$ and 2 $CaCl_2.H_2O$; pH 7.4) for 30 min at 30°C. The slices were then transferred to a room temperature ($\sim$23°C) bath and allowed to recover for an additional hour before further experimentation.

## Light transmittance experiments: intrinsic optical imaging procedure

Following incubation and recovery hemi-sected brain slices were individually transferred to a submerged chamber continuously perfused with carboxygenated aCSF at 30–32°C. Slices were visualized under $\times$4 bright-field using a Zeiss Axioexaminer microscope (Carl Zeiss Microscopy, LLC, White Plains, NY, USA). Intrinsic optical signal (IOS) imaging was then performed as previously described (Anderson & Andrew, 2002; Andrew et al., 2017). In brief, slices were video recorded using a PixelFly QE camera (PCO, Kelheim, Germany) at 0.2 Hz and captured using Axon Imaging Workbench (AIW) (Indec Biosystems, Los Altos, CA, USA). Initial pre-baseline images were obtained for each slice by averaging a series of control images. The percentage change in IOS images was then obtained by subtracting and normalizing this average baseline image ($T_{cont}$) from all subsequent experimental images ($T_{exp}$) (i.e. IOS Percentage Change $= [(T_{exp} - T_{cont})/T_{cont}] \times 100\%$). Resulting images are displayed using a pseudo-colour intensity scale. Regions of interest (ROIs) across the cortical region were selected using the AIW software and saved for off-line analysis. To enhance rigor a negative and positive control were included for each animal. The first slice was used as a negative control slice, with the same ACSF as the incubation bath flowing through the recording chamber and examined for induction of cortical spreading depression (CSD) over a 30-min period. If no CSD was observed within the 30-min period to control aCSF, a positive control was then employed by testing for CSD induction through bath application of an osmolality balanced high potassium (26 mM $K^+$, KCl substituted for equimolar concentrations of NaCl). CSD was observed as a propagating wave of high intensity IOS change across the cortical area as previously described (Anderson & Andrew, 2002). Finally, to ensure longitudinal viability and tissue quality, this negative and positive control procedure was also performed on the last slice of a recording session each day. All animals tested successfully passed negative and positive control testing negating the need for exclusion of any recording sessions. The remaining slices were then bathed in vehicle (0.02% DMSO)- or LEI (0.02% DMSO +

10 µM LEI-106)-containing aCSF and monitored by IOS imaging for changes in IOS values and induction of CSD.

### Laser speckle contrast imaging to assess cortical perfusion

Rats were anaesthetized with 3% isoflurane mixed with breathing air. The depth of anaesthesia was monitored by checking withdrawal and corneal reflexes along with monitoring the oxygenation and cardiorespiratory function (presence of pink mucous membranes, observing chest wall movement, rate and character of respiration) at the beginning and throughout the process. Upon confirmation of deep anaesthesia, rats were moved to a stereotaxic frame (Stoelting Co., Wood Dale, IL, USA) to ensure head immobilization using ear bars. Body temperature was maintained constant at $37 \pm 1°C$ using a rectal probe that provided feedback to a heating pad (RightTemp, Kent Scientific Corp., Torrington, CT, USA). The scalp was shaved and exposed, a midline incision was performed to expose the skull, and the periosteum was gently removed with cotton-tipped applicators. A thinned-skull cranial window was then performed on the parietal bone atop the cerebrum, between the lambda suture and bregma using a micro-drill (Braintree Scientific Inc., Braintree, MA, USA) on both hemispheres (Yang et al., 2010). Care was taken to not thin the sagittal suture, which served as a barrier to prevent fluid from moving between the two hemispheres of the brain. Using this technique, we were able to apply LEI-106 (10 µM) to one hemisphere and vehicle (0.02% DMSO in aCSF) to the other without cross-contamination (see representative traces in Fig. 6). After finalizing the thinned-skull cranial window, isoflurane was lowered to 1.5%, which still provided an acceptable level of anaesthesia monitored as stated above (Constantinides et al., 2011), including observable haemodynamic responses after stimulation (Franceschini et al., 2010). The laser speckle contrast imaging system (PSI-Z; Perimed AB, Järfälla, Sweden) was placed 11–12 cm above the skull, cerebral perfusion was recorded in real-time at a rate at a rate of 5 images/s. Rats were allowed to stabilize for 15 min, then LEI-106 was applied to one hemisphere and vehicle to the other simultaneously (randomized treatment order for each rat), and perfusion was recorded for 5 more minutes. At the end of the recording, rats were euthanized by exsanguination followed by decapitation, and brains were collected for pressure myography experiments. All perfusion data were acquired using the manufacturer's software (PIMsoft v. 1.6; Perimed) and are expressed as perfusion units (PU) (Blackwell et al., 2022). Changes in perfusion with treatments are shown as a percentage increase from pre-treatment (average perfusion 1 min before applying LEI-106 or vehicle).

### Pressure myography of isolated of pial arteries and penetrating arterioles

Pial arteries (posterior communicating artery) and penetrating arterioles were isolated as described previously (Pires et al., 2015; Peters et al., 2022). Briefly, excised brains were placed in a dissection dish filled with ice-cold tissue collection physiological salt solution (PSS, in mM): 140 NaCl, 5 KCl, 2 $MgCl_2$, 10 dextrose, 10 Hepes, pH 7.4, supplemented with 0.05% BSA (BP1600-100, Thermo Fisher Scientific). Posterior communicating arteries were carefully dissected and set aside for experiments. A $5 \times 3 \times 3$ mm rectangular cuboid of tissue around the middle cerebral artery (MCA) was removed, and the superficial cortex separated from the striatum at the level of the corpus callosum. The MCA was then carefully peeled from the cortex to preserve the integrity of branching penetrating arterioles. After isolation, pial arteries and penetrating arterioles were mounted in a custom-made pressure myograph chamber filled with PSS and mounted onto small glass cannulas filled with artificial cerebrospinal fluid (in mM: 124 NaCl, 3 KCl, 2 $MgCl_2$, 1.085 $NaH_2PO_4.2H_2O$, 26 $NaHCO_3$, 1.8 $CaCl_2$, 4 dextrose) and pressurized under isobaric, no-flow conditions. Preparations were moved to a microscope for live imaging using an edge-detection system to identify arterial walls and lumen (IonWizard v7.3 software, IonOptix, Westwood, MA, USA). Warm (37°C), oxygenated (21% $O_2$/5% $CO_2$/balance $N_2$) aCSF was exchanged at a rate of 3–5 ml/min throughout the experiment. Preparations were initially pressurized at 15 mmHg for a 30-min equilibration, then intraluminal pressure was increased to 80 mmHg (pial arteries) or 60 mmHg (penetrating arterioles) to allow the generation of spontaneous myogenic tone. Lumen diameters were recorded in real-time at 15 Hz.

Myogenic reactivity curves were obtained in all preparations in a paired experimental design: one curve in the presence of one treatment (vehicle or LEI-106), followed by washout, then incubation with the other treatment for 5 min (vehicle or LEI-106), followed by another myogenic reactivity curve. Treatment order was randomized for each arteriole to avoid artifacts (50% of preparations started with the vehicle, 50% started with 10 µM LEI-106). Myogenic reactivity curves were performed by increasing intraluminal pressure from 5 to 140 mmHg in 20 mmHg increments; preparations were equilibrated at each pressure for 5 min before the next step (Pires et al., 2017). At the end of these experiments, preparations were bathed in $Ca^{2+}$-free aCSF supplemented with EGTA (2 mM), diltiazem (10 µM) and sodium nitroprusside (100 µM) to record passive diameter at each pressure. Lumen diameter at each pressure was recorded as the average diameter through the last minute of that pressure step. Myogenic tone at each pressure was calculated as: Myo-

genic tone (%) = [1 − (LDT/LDP)] × 100, where LDT is the lumen diameter with myogenic tone and LDP is the passive lumen diameter.

## Statistical analysis

GraphPad Prism 9.5 software (GraphPad Software, Boston, MA, USA) was used for statistical analysis. To determine the numbers needed for each experiment, G.Power3.1 was used for 80% power to detect a 20% effect ($\rho = 0.2$) when $\alpha = 0.05$. The data were expressed as means ± SD unless otherwise stated. Groups were assessed for normality and sphericity and then compared by unpaired Student's *t* test, one-way ANOVA with Tukey's *post-hoc* test, or repeated measures two-way ANOVA (myogenic reactivity experiments) with Šídák's correction, as appropriate and indicated. Differences were considered significant if $P \leq 0.05$.

## Results

### The blood–brain barrier was compromised *in vivo* after pharmacological inhibition of DAGLα

Previously, changes in the barrier integrity of cultured brain endothelial cells after 2-AG depletion were observed *in vitro* (Levine et al., 2024). Our previous results also showed that pharmacological inhibition of DAGLα by intraperitoneal injection of LEI-106 induced headache-like behaviours of periorbital allodynia, photophobia and anxiety in Sprague–Dawley rats (Levine et al., 2021). Notably, female animals showed significantly higher sensitivity to DAGLα inhibition, compared to male animals in our previous work, and therefore female Sprague–Dawley rats were utilized in the current paper. It is reported that the integrity of the BBB is transiently disrupted in headache-related disorders like migraine or traumatic brain injury (Wiggers et al., 2022). To connect these observations, the first sets of experiments were designed to test the impact of pharmacological blockade of DAGLα on *in vivo* BBB permeability.

Female rats underwent carotid cannulation and *in situ* brain perfusion with [$^{14}$C]sucrose to determine paracellular permeability of the BBB 90 min following LEI-106 (40 mg/kg, I.P.) or vehicle (1:1:8, DMSO–Tween 80–0.9% saline) administration (Fig. 1*A*). After 15 min of carotid perfusion, brains were harvested, and vasculature was removed (i.e. capillary depletion). Tissue samples of cortex, PAG and medulla were analysed for extravasation of [$^{14}$C]sucrose. The ratio between tissue radioactivity and perfusate radioactivity (i.e. RBR) was calculated for

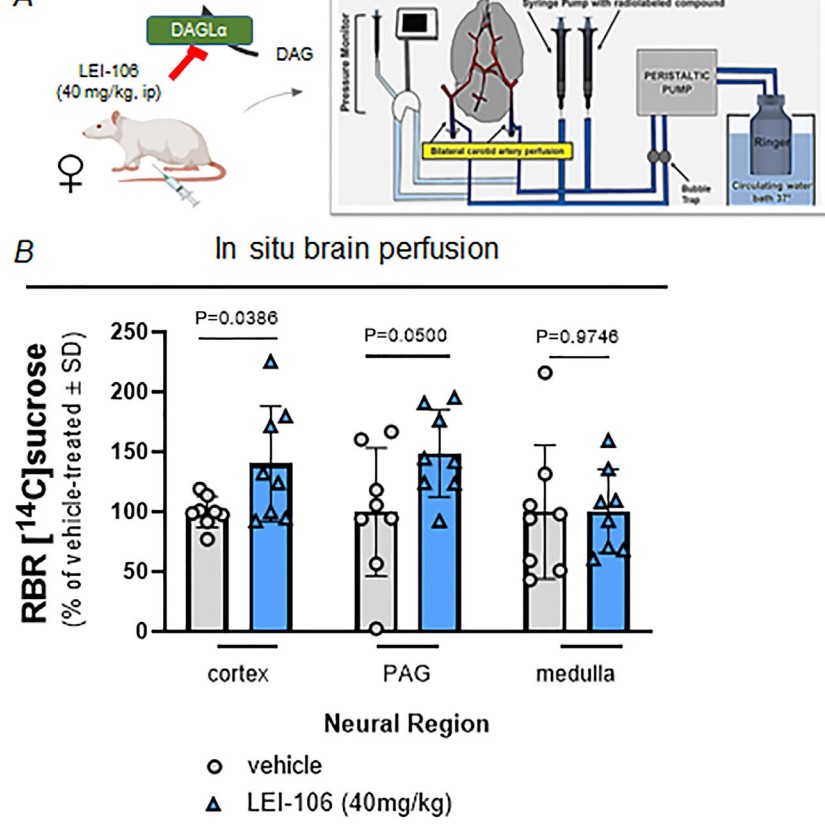

**Figure 1. The pharmacological blockade of DAGLα induced breaches in BBB integrity**
Female Sprague–Dawley rats were injected with the DAGLα inhibitor LEI-106 (40 mg/kg, I.P.). The carotid cannulation and perfusion were performed using [$^{14}$C]sucrose at 90 min after LEI-106 administration. Following a 15 min perfusion, the cortices, PAG and medulla were isolated, and then the radioactivity was measured with a liquid scintillation counter. *A*, schematic flow of experiment (Created with BioRender.com). *B*, significant increase of RBR (ratio between tissue radioactivity and perfusate radioactivity) was observed in the cortex and PAG samples after LEI-106 administration as assessed by unpaired *t* test, suggesting compromised BBB integrity and paracellular leak of [$^{14}$C]sucrose. The LEI-106 treatment did not cause significant changes of RBR in medulla samples as assessed by unpaired *t* test, indicating the region-specific effect of DAGLα on BBB integrity. All data represent the percentage of vehicle-treated ± SD (*n* = 8 in each group). [Colour figure can be viewed at wileyonlinelibrary.com]

each brain region. Increased RBR values were observed in the cortex (Fig. 1*B*) (*P* = 0.0386, *t*(14) = 2.283 as assessed by unpaired *t* test, *n* = 8/group) and PAG (*P* = 0.0500, *t*(14) = 2.142 as assessed by unpaired *t* test, *n* = 8/group), from LEI-106-treated animals, indicating breaches in BBB paracellular integrity. No significant change of RBR was measured in medulla samples, which also contain the trigeminal brainstem complex (Fig. 1*B*; *P* = 0.9746, *t*(14) = 0.0324 as assessed by unpaired *t* test, *n* = 8/group). Together, these data suggest that blocking DAGL$\alpha$ promotes paracellular leak of the BBB in a region-dependent manner.

### DAGL$\alpha$ blockade reduced the expression of the tight junction protein ZO-1

To examine the changes in expression of tight junction proteins as a possible molecular mechanism of BBB opening caused by DAGL$\alpha$ inhibition, female Sprague–Dawley rats were treated with the DAGL$\alpha$ inhibitor LEI-106 (40 mg/kg, I.P.), followed by tissue (cortex and PAG) harvest at 90 min after LEI-106 administration. Samples were subjected to western immunoblotting to measure total protein detection of ZO-1, VE-cadherin and claudin-5 as compared to the loading control of $\alpha$-tubulin (Fig. 2*A*). Administration of LEI-106 significantly decreased the detection of ZO-1 in cortex samples, compared to vehicle control (Fig. 2*B*; *P* = 0.0302, *t*(10) = 2.524 as assessed by unpaired *t* test, *n* = 5–6/group). No significant changes in the expression of VE-cadherin and claudin-5 were detected in cortex samples after LEI-106 injection (Fig. 2*B*; VE-cadherin: *P* = 0.9307, *t*(10) = 0.08919; claudin-5: *P* = 0.2330, *t*(9) = 1.279 as assessed by unpaired *t* test, *n* = 5–6/group). Within the PAG, systemic dosing with LEI-106 did not significantly change the detection of ZO-1, VE-cadherin or claudin-5 as compared to vehicle control (Fig. 2*C*; ZO-1 *P* = 0.9835, *t*(9) = 0.0212; VE-cadherin: *P* = 0.6741, *t*(9) = 0.04346; claudin-5: *P* = 0.2849, *t*(9) = 1.137 as assessed by unpaired *t* test, *n* = 5–6/group). These results indicate that the loss in the cortical detection of ZO-1 may play a role in BBB paracellular leak after DAGL$\alpha$ inhibition, however the molecular mechanism of BBB disruption within PAG remains elusive.

### The pharmacological blockade of DAGL$\alpha$-induced region-dependent changes of endocannabinoid levels

Next, we evaluated how the observed changes in the integrity of the BBB associated to regional endocannabinoid levels in the cortex and PAG; serum endocannabinoid levels after systemic injection of LEI-106 (40 mg/kg) in female animals were also assessed (Fig. 3*A*). Systemic blockade of DAGL$\alpha$ significantly reduced the

2-AG level in cortex samples, without altering AEA level as compared to vehicle control (Fig. 3*B* and *C*, respectively; 2-AG: *P* = 0.0280, *t*(9) = 2.617; AEA: *P* = 0.7823, *t*(9) = 0.2847 as assessed by unpaired *t* test, *n* = 5–6 in each group). No significant differences in the level of either 2-AG or AEA was detected in PAG after LEI-106 injection compared to vehicle control (Fig. 3*D* and *E*, respectively; 2-AG: *P* = 0.9743, *t*(10) = 0.0331; AEA: *P* = 0.7471, *t*(10) = 0.3316 as assessed by unpaired *t* test, *n* = 6 in each group). The serum level of 2-AG was significantly increased 90 min after LEI-106 administration compared to vehicle control (Fig. 3*F*; *P* = 0.0172, *t*(8) = 2.997 as assessed by unpaired *t* test, *n* = 5 in each group). No change was observed in the serum level of AEA after DAGL$\alpha$ blockade (Fig. 3*G*; *P* = 0.6614, *t*(10) = 0.4514 as assessed by unpaired *t* test, *n* = 6 in each group). These data indicate that LEI-106 (40 mg/kg) reduced CNS 2-AG levels in a region-selective manner that coupled to increases in serum detection.

### DAGL$\alpha$ inhibition with LEI-106 has limited impact on neuronal activity in the cortex

Several research papers have confirmed that CSD induces region-dependent changes in BBB integrity (Cottier et al., 2018; Gursoy-Ozdemir et al., 2004; Olah et al., 2013). Our previous results have indicated that the ECS is also comprised, mainly via reduced levels of 2-AG, after the induction of CSD in female Sprague–Dawley rats (Liktor-Busa et al. 2023). To examine if the depleted 2-AG levels were related to possible CSD and changes in neuronal activity, tissue samples were harvested 90 min after systemic injection of DAGL$\alpha$ inhibitor and subjected to western blot to detect NPAS4 (Fig. 4*A*). The neuronal PAS domain protein 4 (NPAS4), encoded by an early immediate gene implicated in regulation of synaptic activity and plasticity (Lin et al., 2008), was recently identified as a marker for brain regions engaged by CSD propagation (Yoshida et al., 2015). Systemic injection of LEI-106 (40 mg/kg, I.P.) statistically increased NPAS4 detection in the cortex, but not the PAG, 90 min after injection of DAGL$\alpha$ inhibitor (Fig. 4*B* and *C*, respectively; cortex: *P* = 0.011, *t*(10) = 3.112; PAG: *P* = 0.2048, *t*(10) = 1.356 as assessed by unpaired *t* test, *n* = 6/group) suggesting a regionally selective increase in neuronal activation.

Previous results showed that cortical cFos expression was elevated 90 min after CSD induction in female Sprague–Dawley rats (Liktor-Busa et al., 2020), and moreover, elevation in cFos expression was observed in NPAS4 KO mice during epileptogenesis (Shan et al., 2018). Therefore, cortex and PAG samples were also probed for cFos protein levels. LEI-106 (40 mg/kg) did not significantly change the cFos detection in cortex or PAG

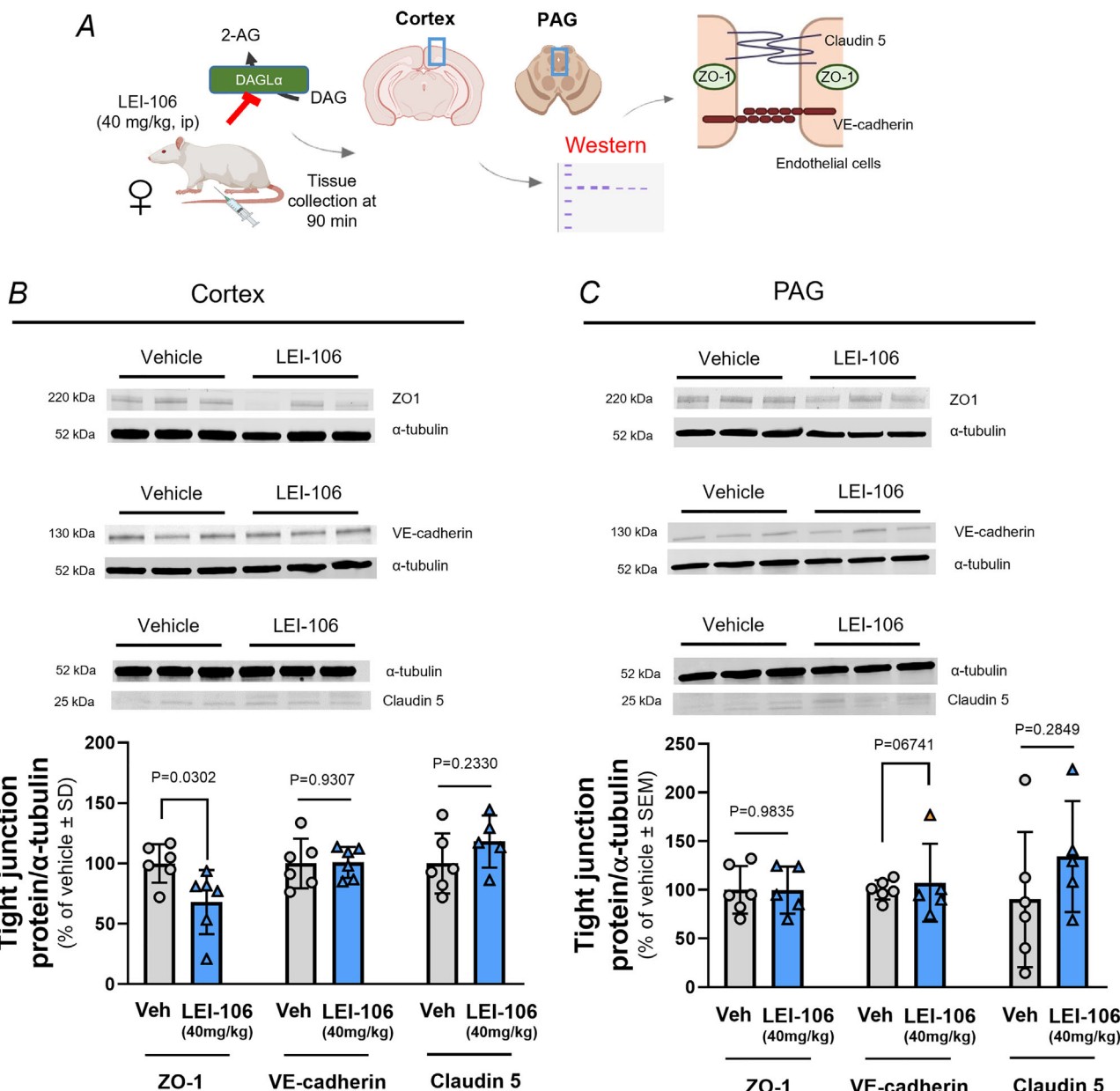

**Figure 2. The pharmacological inhibition of DAGLα reduced the expression of ZO-1 in the cortex**

Female Sprague–Dawley rats were treated with the DAGLα inhibitor LEI-106 (40 mg/kg, I.P.). Tissue samples, including cortex and PAG, were harvested at 90 min after LEI-106 administration. Samples were subjected to western immunoblotting to measure the expression of tight junction proteins ZO-1, VE-cadherin and claudin-5. *A*, schema of experiment (Created with BioRender.com). *B*, representative image showing the expression of ZO-1, VE-cadherin and claudin-5 with α-tubulin as loading control in cortex samples. The administration of DAGLα inhibitor significantly decreased the detection of ZO-1 in cortex samples, compared to vehicle control (LEI-106 *vs.* vehicle: $P = 0.0302$, $t(10) = 2.524$, as assessed by unpaired *t* test). No significant changes in the expression of VE-cadherin and claudin-5 were detected in cortex samples after LEI-106 injection (VE-cadherin: LEI-106 *vs.* vehicle: $P = 0.9307$, $t(10) = 0.08919$; claudin-5: LEI-106 *vs.* vehicle: $P = 0.2330$, $t(9) = 1.279$ as assessed by unpaired *t* test). Data are shown as a percentage of vehicle control ± SD ($n = 5$–6/group). *C*, representative western blot displaying the expression of ZO-1, VE-cadherin and claudin-5 in PAG samples. α-Tubulin was used as a loading control. VE-Cadherin and claudin-5 were probed for on the same gel. α-Tubulin (680 filter) was imaged at a low intensity with VE-cadherin (680). To view claudin-5, the 680 filter was reimaged at a higher intensity. To reflect the difference in intensity used for normalization, the α-tubulin band is shown with each target protein at

the corresponding imaged intensity. The injection of LEI-106 did not induce significant change in the detection of any tight junction protein assessed in PAG as compared to vehicle control (ZO-1: LEI-106 *vs.* vehicle: $P = 0.9835$, $t(9) = 0.0212$, VE-cadherin: LEI-106 *vs.* vehicle: $P = 0.6741$, $t(9) = 0.04346$, claudin-5: LEI-106 *vs.* vehicle: $P = 0.2849$, $t(9) = 1.137$, as assessed by unpaired *t* test). Data are shown as a percentage of vehicle control ± SD ($n = 5$–6/group). Orange denoted statistically identified outlier, ROUT 10%. [Colour figure can be viewed at wileyonlinelibrary.com]

samples as compared to vehicle control (Fig. 4*D* and *E*, respectively; cortex: $P = 0.8058$, $t(9) = 0.2532$; PAG: $P = 0.1297$, $t(9) = 0.1668$ as assessed by unpaired *t* test, $n = 5$–6 in each group).

As a follow-up after western immunoblotting, qPCR experiments were conducted. Cortex samples harvested 90 min after LEI-106 injection (40 mg/kg, I.P.) were sub-

jected to qPCR detection of mRNA levels of *Npas4* and *cFos*. Neither the relative expression of *Npas4* mRNA nor that of *cFos* mRNA in the cortex was significantly influenced by the administration of DAGLα inhibitor as compared to vehicle control (Fig. 4*F* and *G*, respectively; *Npas4*: $P = 0.9141$, $t(10) = 0.1107$; *cFos*: $P = 0.3228$, $t(10) = 1.040$ as assessed by unpaired *t* test, $n = 6$ in each group).

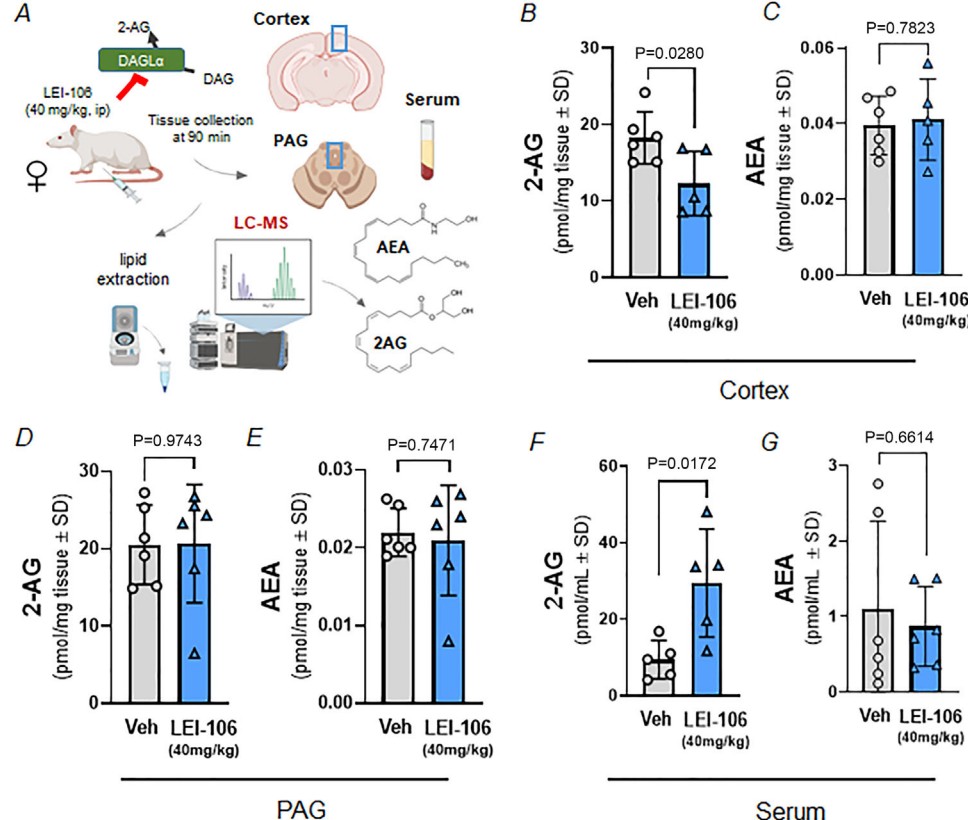

**Figure 3. The inhibition of DAGLα alters endocannabinoid composition in a region-dependent manner**
Female Sprague–Dawley rats were injected with the selective DAGLα inhibitor LEI-106 (40 mg/kg, I.P.). Tissue (cortex and PAG) and serum samples were harvested 90 min after the administration of LEI-106, then subjected to LC-MS to quantify the level of endocannabinoids, 2-AG and AEA. *A*, schema of experimental setting (Created with BioRender.com). *B*, the level of 2-AG in cortex was significantly reduced after LEI-106 administration, compared to vehicle control, as assessed by unpaired *t* test. Data are shown as means ± SD in pmol/mg tissue ($n = 5$–6 in each group). *C*, the injection of LEI-106 did not cause significant changes in the cortex level of AEA compared to control, as assessed by unpaired *t* test. Data are shown as mean ± SD in pmol/mg tissue ($n = 5$–6 in each group). *D*, no significant difference was observed in the level of 2-AG in PAG after LEI-106 injection, as assessed by unpaired *t* test. Data are shown as means ± SD in pmol/mg tissue ($n = 6$ in each group). *E*, the administration of DAGLα inhibitor did not induce significant change in AEA level of PAG, as assessed by unpaired *t* test. Data are shown as means ± SD in pmol/mg tissue ($n = 6$ in each group). *F*, the serum level of 2-AG was significantly increased 90 min after LEI-106 administration compared to vehicle control, as assessed by unpaired *t* test. Data are shown as means ± SD in pmol/ml ($n = 5$ in each group). *G*, LEI-106 administration did not significantly influence the serum level of AEA as compared to vehicle control, as assessed by unpaired *t* test. Data are shown as means ± SD in pmol/ml ($n = 6$ in each group). [Colour figure can be viewed at wileyonlinelibrary.com]

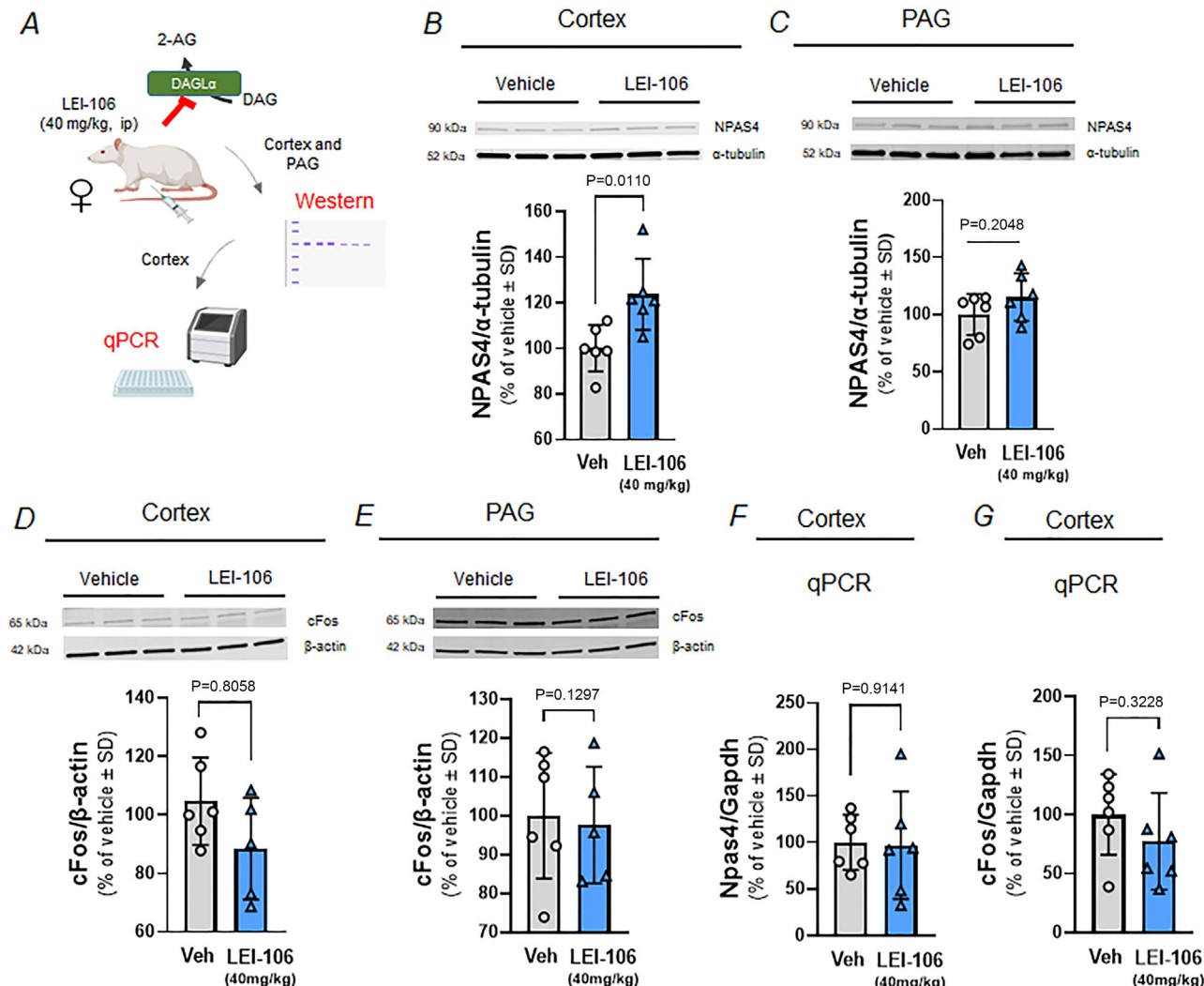

**Figure 4. The inhibition of DAGLα increased the detection of NPAS4 in the cortex**

Female Sprague–Dawley rats were injected with DAGLα inhibitor, LEI-106 (40 mg/kg, I.P.) 90 min prior tissue harvest. Cortex and PAG samples were harvested and then subjected to western blotting to detect NPAS4 and cFos protein. In a separate experiment, cortex samples were subjected to a qPCR experiment to quantify the mRNA level of *NPAS4* and *cFos* genes. *A*, schematic outline of the experimental setting (Created with BioRender.com). *B* and *C*, representative western blot image showing NPAS4 in cortex (*B*) and PAG (*C*) samples. α-Tubulin was used as a loading control. The blockade of DAGLα induced a significant elevation in the detection of NPAS4 in the cortex as compared to vehicle control, as assessed by unpaired *t* test, suggesting an increase in neuronal activation after LEI-106 administration. Western experiments showed no significant change in the detection of NPAS4 in PAG as compared to vehicle control, as assessed by unpaired *t* test. All data are shown as the percentage of vehicle-treated ± SD (*n* = 6 in each group). *D* and *E*, representative western blot image showing cFos in cortex (*D*) and PAG (*E*) samples. β-Actin was used as a loading control. *D*, no significant change in the detection of cFos was observed in cortex samples after DAGLα inhibition as compared to vehicle control, as assessed by unpaired *t* test. All data are shown as the percentage of vehicle-treated ± SD (*n* = 5–6 in each group). *E*, the administration of LEI-106 did not significantly modify the detection of cFos protein in PAG as compared to vehicle control, as assessed by unpaired *t* test. All data are shown as the percentage of vehicle-treated ± SD (*n* = 5–6 in each group). *F*, the expression of *Npas4* mRNA was not significantly changed in cortex 90 min after LEI-106 administration as compared to vehicle control, as assessed by unpaired *t* test. All data show the relative expression of *Npas4* as a percentage of vehicle-treated ± SD (*n* = 6 in each group). *Gapdh* was used as a reference gene. *G*, qPCR experiment revealed no significant difference in the expression of *cFos* mRNA in the cortex after the inhibition of DAGLα as compared to vehicle control, as assessed by unpaired *t* test. All data show the relative expression of *cFos* as the percentage of vehicle-treated ± SD (*n* = 6 in each group). *Gapdh* was used as a reference gene. [Colour figure can be viewed at wileyonlinelibrary.com]

To test the possible presence of CSD events after 2-AG depletion, naive female Sprague–Dawley rats were sacrificed under deep isoflurane anaesthesia, brains were harvested and 350 μm slices were taken in the region of the visual cortex (V1). Slices were placed into a submerged chamber and changes in IOS imaging were monitored as depicted in the schematic representation in Fig. 5*A*. We monitored for induction of CSD as a propagating wave of increased light transmission as previously described (Anderson & Andrew, 2002). Individual brain slices were then treated with either vehicle (0.02% DMSO in aCSF; n = 35 slices) or LEI-106 (10 μM LEI-106 in 0.02% DMSO-containing aCSF; n = 41) for 30 min and changes in IOS activity and potential development of CSD monitored (Fig. 5*B*). ROIs across the cortical layers were defined as previously described (Anderson & Andrew,

2002) and IOS quantified. We first examined average IOS intensity across all cortical layers at 10-min intervals during vehicle or LEI-106 treatment. The treatment of slices with 10 μM of LEI-106 did not significantly increase peak IOS intensity when compared to vehicle treatment at 30 s (Vehicle = −1.075 ± 0.42%; LEI = −0.634 ± 0.35%), 10 min (Vehicle = −0.250 ± 2.5%; LEI-106 = 2.047 ± 1.78%), 20 min (Vehicle = 1.585 ± 2.96%; LEI-106 = 3.005 ± 2.02%) or 30 min of treatment (Vehicle = 2.338 ± 2.26%; LEI-106 = 2.098 ± 2.00%) (Fig. 4*C*). As a result, we determined that no CSD events were detected in any animals across any treatments (i.e. LEI-106 or vehicle) (LEI-106 *vs.* vehicle, 30 s: $P = 0.9931$, 10 min: $P = 0.2204$, 20 min: $P = 0.6639$, 30 min: $P = 0.9993$ as assessed by two-way ANOVA with Šídák's *post hoc* test, $F_{(3,36)} = 0.8810$). Of note, these vehicles and LEI-106 intensity

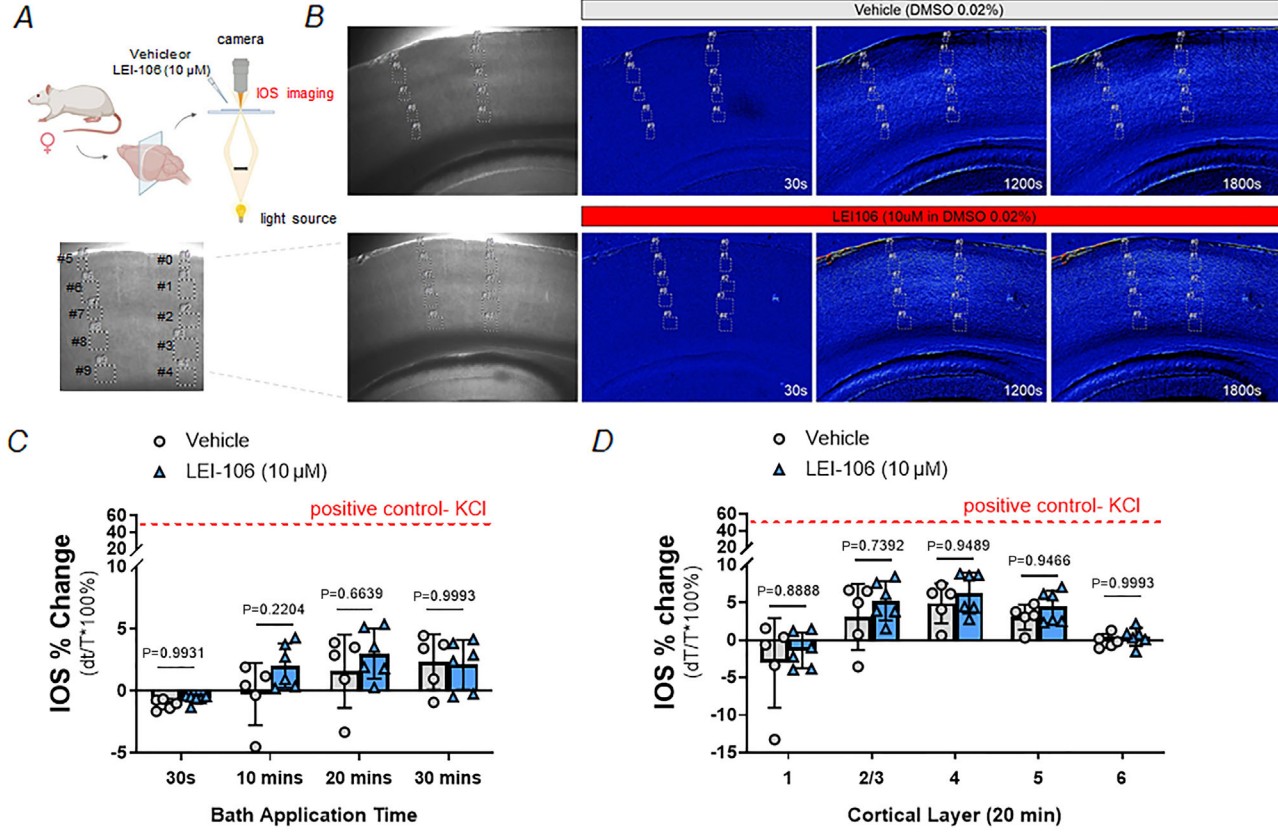

**Figure 5. The inhibition of DAGLα did not induce CSD events**
Naive female Sprague–Dawley rats were sacrificed under deep isoflurane anaesthesia, brains were harvested and 350 μm slices taken in the region of the visual cortex (V1), followed by the measurement of light transmittance as described in 'Methods'. *A*, schematic outline of the experimental setting (Created with BioRender.com). *B*, representative images of cortical slices treated with either vehicle (0.02% DMSO in aCSF) or LEI-106 (10 μM LEI-106 in 0.02% DMSO-containing aCSF). *C*, the application of LEI-106 (10 μM) did not cause significant change in IOS as compared to vehicle control over application time, as assessed by two-way ANOVA with Šídák's *post hoc* test, indicating the absence of cortical spreading depression. All data are expressed as percentage of total cortical average across time ± SD (n = 5–6/group). *D*, the inhibition of DAGLα by LEI-106 application did not cause significant change in IOS as compared to vehicle control in any cortical layers as assessed by two-way ANOVA with Šídák's *post hoc* test. All data are expressed as percentage of total average at 20 min time point ± SD (n = 5–6/group). [Colour figure can be viewed at wileyonlinelibrary.com]

values were substantially smaller than the 40% increase in IOS observed during the peak of CSD induced during our positive control testing by 26 mM K$^+$ (Fig. 4*C* and *D*, red dotted line). To examine differences in cell layer specific effects of LEI-106 treatment in IOS intensity across five cortical regions (layer 1, layer 2/3, layer 4, layer 5 and layer 6), IOS values in slices treated for 20 min with either vehicle or LEI-106 were determined. As expected based on our previous results, some differences in IOS intensity were observed across cortical layers (Anderson & Andrew, 2002; Andrew et al., 2017) with a trend towards higher IOS intensity values in the middle cortical layers (i.e. Layer 1: Vehicle = −3.050/−5.98%; LEI-106 = −1.377 ± 2.37%; Layer 2/3: Vehicle = 3.089 ± 4.41%; LEI-106 = 5.242 ± 2.61%; Layer 4: Vehicle = 4.91 ± 2.67%; LEI-106 = 6.281 ± 2.72%; Layer 5: Vehicle = 3.059 ± 1.67%; LEI-106 = 4.444 ± 2.16%; Layer 6: Vehicle = −0.083 ± 0.92%; LEI-106 = 0.448 ± 1.21%; Fig. 4*D*). No significant effect of LEI-106 treatment was observed across any cortical layer when compared to vehicle-treated slices (LEI-106 *vs.* vehicle, 1: $P = 0.8888$, 2/3: $P = 0.7392$, 4: $P = 0.9489$, 5: $P = 0.9466$, 6: $P = 0.9993$ as assessed by two-way ANOVA with Šídák *post hoc* test, $F_{(4,45)} = 0.1087$). Overall, these data suggest that LEI-106 (10 µM) fails to directly induce CSD or significant changes in cortical excitability using the proxy measurement of changes in IOS intensity. Together, the *Npas4*, *cFos* and CSD slice data suggest that systemic administration of the DAGLα inhibitor LEI-106 can change CNS activity *in vivo* in a limited manner, but this change in activity does not correspond to CSD induction *in vitro*.

## LEI-106 increases pressure-induced contractility of penetrating arterioles, without affecting pial arteries

Changes in cerebral haemodynamic regulation can affect the BBB, as physiological shear stress is important to maintain the structural integrity of tight junctions in the microvascular endothelium (Cucullo et al., 2011). Shear stress is a consequence of the friction caused by fluid moving on top of endothelial cells, which creates a tangential force that disturbs the endothelium. Further, levels of shear stress can be altered by vascular diameter, and thus pressure myography experiments were performed to assess if acute 2-AG depletion changes physiological cerebral vascular reactivity, thus potentially affecting shear stress. We observed that acute exposure of isolated, pressurized pial arteries to LEI-106 (10 µM) had no effect on pressure-induced contractility (Fig. 6*A* and *B*; LEI-106 *vs.* vehicle: $P > 0.05$, as assessed by mixed-model two-way ANOVA with Šídák's *post hoc* test, $F_{(7,42)} = 1.801$, $n = 4$ arteries from four different rats). However, LEI-106 induced a significant increase in pressure-induced constriction of penetrating arterioles in

the physiological range of 40–80 mmHg when compared to vehicle controls (Fig. 6*C* and *D*; LEI-106 *vs.* vehicle: 40 mmHg: $P = 0.0035$, 60 mmHg: $P = 0.0181$, 80 mmHg: $P = 0.0196$, as assessed by mixed-model two-way ANOVA with Šídák's *post hoc* test, $F_{(7,96)} = 0.6191$, $n = 7$ penetrating arterioles from six different rats). These data suggest that DAGLα inhibition increases vascular resistance in the cerebral microcirculation, potentially leading to an increase in shear stress.

## DAGLα inhibition increases superficial cortical perfusion

We next assessed the *in vivo* cerebral haemodynamic consequences of acute DAGLα inhibition on the cortical circulation using laser speckle contrast imaging. We observed that acute application of LEI-106 (10 µM) on top of the thinned-skull cranial window in one hemisphere caused a significant increase in superficial cortical perfusion, without affecting perfusion to the contralateral cortex (simultaneously exposed to vehicle, Fig. 6*E–G*). As observed in the representative traces (Fig. 6*F*), the increase in perfusion was nearly immediate and sustained over time, returning to baseline levels about 10 min after washout. The significant increase in perfusion caused by LEI-106 application can also be seen in the summary graph (Fig. 6*G*; LEI-106 *vs.* vehicle: $P = 0.0001$, $t(12) = 5.453$ as assessed by unpaired *t* test, $n = 7$). Application of vehicle did not affect cortical perfusion (Fig. 6*E* and *F*). Together with pressure myography data, our findings suggest that acute reductions in 2-AG may induce a generalized constriction of penetrating arterioles, resulting in an increase in superficial cortical perfusion.

## Discussion

The protective role of the ECS on the BBB during injury has been proposed (Hagan et al., 2022; Vendel E & de Lange, 2014), but for the first time, this study investigates the *in vivo* effects of endocannabinoid depletion on the integrity of intact, non-injured BBB. We found that systemic injection of the DAGLα inhibitor LEI-106 induced region-dependent opening of the BBB in female Sprague–Dawley rats, accompanied by reduced 2-AG. The pharmacological blockade of DAGLα reduced the cortical expression of the tight junction protein ZO-1, which is one possible molecular mechanism underlying the observed BBB breaches. Light transmittance experiments showed that CSD was not induced after LEI-106 application, despite some alterations in neuronal activity as determined by NPAS4 detection. Lastly, our haemodynamic experiments suggest that reduced levels of cortical 2-AG alter cerebrovascular regulation, observed as an increase in arteriolar pressure-induced

constriction and accumulation of blood in the superficial cortical circulation. Overall, these results suggest changes in BBB integrity, neural activity and microvascular reactivity independent of CSD induction after DAGL$\alpha$ inhibition.

In our previous paper, we showed that intraperitoneal administration of the DAGL$\alpha$ inhibitor LEI-106 induced headache-like symptoms, including cutaneous allodynia at the cephalic site, and spontaneous pain behaviour, like

head-pressing and photophobia (Levine et al., 2021). It is well-known that headache disorders, like certain forms of migraine or traumatic brain injury, are associated with changes in the permeability of the BBB, leading to worse outcomes (Wiggers et al., 2022). In the current paper, we investigated if similar changes in the BBB can be detected in the headache model induced by DAGL$\alpha$ inhibition. Brain perfusion of [$^{14}$C]sucrose as a marker of BBB paracellular permeability (Cottier et al., 2018) revealed

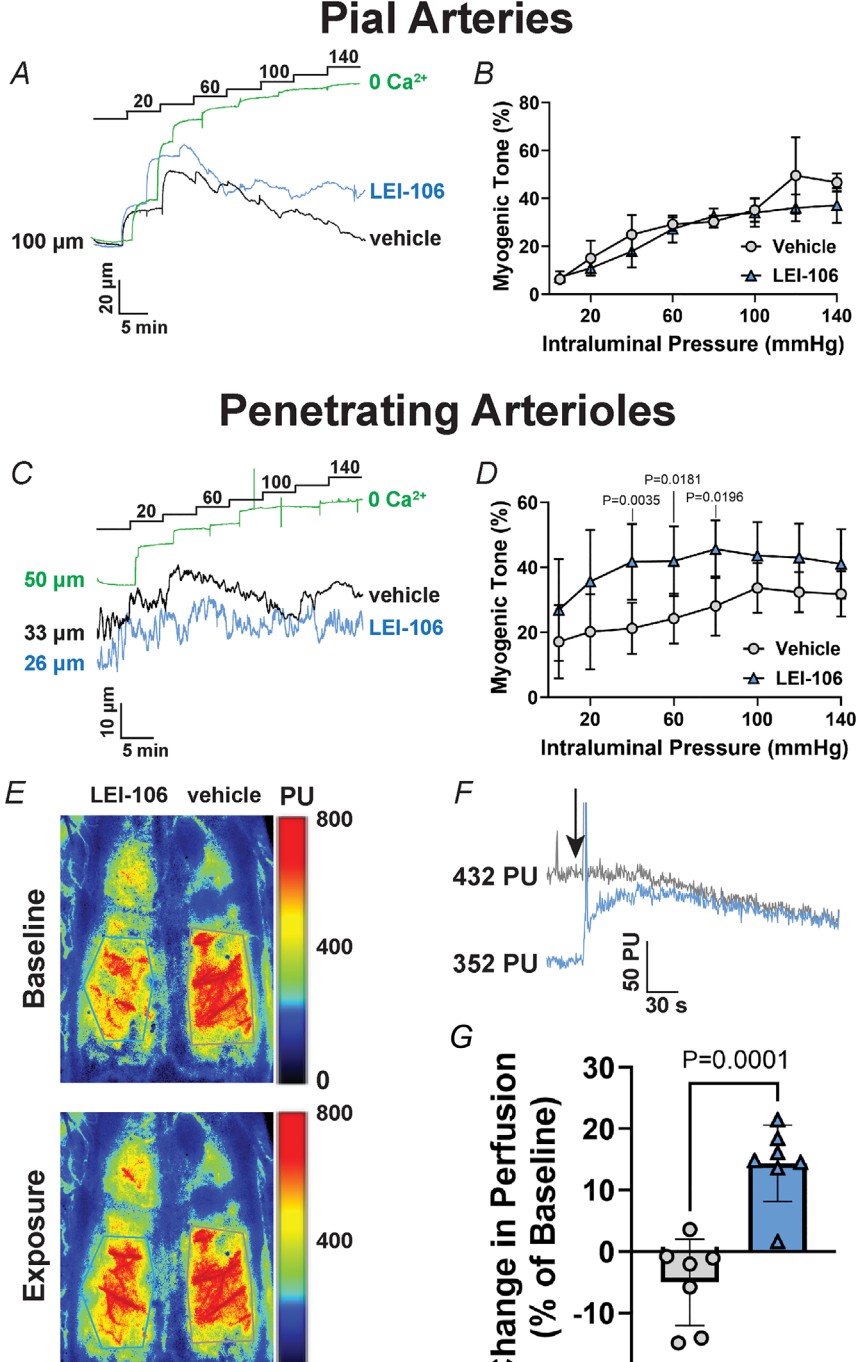

**Figure 6. Effects of acute DAGL$\alpha$ inhibition on cerebral microvascular myogenic reactivity and cortical perfusion**

*A* and *B*, application of the DAGL$\alpha$ inhibitor LEI-106 (10 µM) to the superfusion bath had no effect on myogenic reactivity of isolated, pressurized pial arteries from Sprague–Dawley female rats, as seen in the representative traces (*A*) and summary data (*B*). Data are shown as a percentage of myogenic tone ± SD (*n* = 4 arteries from four different rats) and were analysed by mixed-model two-way ANOVA with Šídák's correction. *C* and *D*, incubation of isolated, pressurized penetrating arterioles with LEI-106 (10 µM) significantly increased myogenic reactivity to stepwise increases in intraluminal pressure, as observed in the representative traces (*C*) and summary data (*D*). Data were analysed by mixed-model two-way ANOVA with Šídák's correction for multiple comparisons. All data are expressed as a percentage of myogenic tone ± SD (*n* = 7 penetrating arterioles from six different rats). *E–G*, LEI-106 (10 µM) caused an acute increase in superficial cortical perfusion when applied on top of the thinned-skull cranial window (left hemisphere in *E*, blue trace in *F*) when compared to simultaneous application of vehicle (0.02% DMSO in aCSF, right hemisphere in *E*, grey trace in *F*). The traces in *F* show that there was no cross-contamination between the different hemispheres (the sagittal suture was not thinned, forming a barrier separating the right and left hemispheres). The arrow in *F* indicates when LEI-106 was applied to the thinned-skull cranial window. *G*, the significant increase in perfusion can be seen in the summary graph and it was assessed by unpaired *t* test. All data are shown as the percentage of baseline ± SD (*n* = 7 female Sprague–Dawley rats). [Colour figure can be viewed at wileyonlinelibrary.com]

that the pharmacological manipulation of DAGLα by systemic injection of LEI-106 caused breaches of the BBB in cortex and PAG, but not in medulla in female Sprague–Dawley rats. Interestingly, the opening of the BBB only in the cortex, but not in PAG, was associated with reduced levels of 2-AG after administration of LEI-106 at a dosage of 40 mg/kg. Of note, systemic injection of LEI-106 at a 20 mg/kg dose significantly decreased 2-AG levels in both brain regions in prior experiments (Levine et al., 2021). Serum 2-AG level was elevated after DAGLα inhibition, which suggests a possible compensatory mechanism in the vasculature. The circulating endocannabinoids can originate from multiple organs, including the brain, muscle, adipose tissue and circulating blood cells (Hillard, 2018). The β isoform of DAGL is the main enzyme responsible for the production of 2-AG in non-neural tissue (Reisenberg et al., 2012), and therefore the serum level of 2-AG might be independent from the activity of DAGLα. Results published by Shonesy and co-workers support this hypothesis, since the normal peripheral 2-AG level was found in DAGLα KO mice, but the brain 2-AG levels were robustly reduced (Shonesy et al., 2014).

There are very few papers available discussing the correlation between serum and cerebrospinal levels of endocannabinoids. A weak, but positive correlation was found between serum and cerebrospinal fluid levels of 2-AG in multiple sclerosis patients (Meier et al., 2024). The level of 2-AG in cerebrospinal fluid was not assessed in this study, but the elevated serum 2-AG level might induce increased 2-AG concentration in the cerebrospinal fluid, which may have an impact on the 2-AG level in the PAG as the cerebral aqueduct is surrounded by the PAG. This is just one possible explanation for the unchanged 2-AG level in the PAG after a higher dose of DAGLα blockade. Other explanations include the possible off-target effects of LEI-106 or the different distribution of DAGLα throughout the CNS. It is known that LEI-106 has off-targets, like α/β-hydrolase domain-containing 6 (ABHD6), which is an enzyme partially responsible for the degradation of 2-AG. The $IC_{50}$ value of LEI-106 for DAGLα (18 nM) is much lower than its value for ABHD6 (0.8 μM) (Janssen et al., 2014), but the higher dose of LEI-106 might mitigate this difference. Our previous data indicate that the administration of LEI-106 has a region-specific impact on the 2-AG levels, and no reduction in the level of 2-AG was found in trigeminal ganglion or trigeminal nucleus caudalis after LEI-106 injection (Levine et al., 2021), suggesting different distribution of DAGLα within the CNS. This hypothesis is supported by literature evidence as well (Suárez et al., 2011). Additional studies beyond the scope of the current work are warranted to elucidate these mechanisms.

Our previous results have indicated that the blockade of DAGLα increases the permeability of cultured brain endothelial cells (bEnd.3), induces fragmentation of the tight junction protein VE-cadherin and decreases the expression of ZO-1 (Levine et al., 2024). The current study confirmed that 2-AG depletion can cause breaches in the BBB *in vivo* through dysregulation of tight junctions. Reduced expression of ZO-1 protein after the blockade of DAGLα was also observed *in vivo*, but the disruption of VE-cadherin after administration of DAGLα inhibitor was not detected in the present study. Several papers have described the protective role of 2-AG signalling on the BBB by administration of exogenous 2-AG or by inhibition of 2-AG degradation routes in mouse models of traumatic brain injury, focal photothrombotic ischaemic insult and after closed head injury (Panikashvili et al. 2006; Piro et al., 2018; Tchantchou & Zhang, 2013). However, the mechanisms of this effect at the level of neurovascular unit have not yet been delineated. A study published by Yang et al. (2016) showed that a cannabinoid-2 receptor agonist, JWH-015, reduces blood–spinal cord barrier permeability during ischaemia–reperfusion injury and induces changes in the distribution of ZO-1, suggesting a possible connection between the ECS and the tight junction protein ZO-1 (Yang et al., 2016). Our data align well with those findings, indicating a direct relation between ZO-1 protein and 2-AG signalling. The level of 2-AG in PAG was not significantly reduced after LEI-106 administration and no change in ZO-1 expression was found. These results suggest that the BBB opening in PAG after LEI-106 administration might be independent from endocannabinoid signalling or caused by off-target effects.

Immediate early genes are transcribed within minutes after stimulation and play roles in regulating gene transcription; in neurons, activation of immediate early genes reflects the activity of neuronal circuits (Okuno, 2011). NPAS4 is an activity-dependent, immediate early gene and transcription factor in neurons that plays a role in maintaining excitatory–inhibitory balance within circuits and is suggested to serve a neuroprotective role during an insult (Fu et al., 2020). As such, a recent paper has identified NPAS4 as a marker of the extent of spread during CSD where elevated *Npas4* mRNA expression coincides with the area of depolarization shift in the ipsilateral dorsal cortex caused by application of KCl (Yoshida et al., 2015). Our data show increased detection of NPAS4 protein in cortex samples after DAGLα blockade without altering the mRNA level of *Npas4*, suggesting that loss of 2-AG signalling disrupts the excitatory–inhibitory balance in this region. Interestingly, no change at the level of *Npas4* mRNA was detected, but change at NPAS4 protein level was observed after DAGLα blockade. Since only one time point was assessed after LEI-106 administration, there is a possibility that transcriptional changes might occur at earlier time points. Further experiments would be required to clarify this question.

In addition to NPAS4, our previous data indicated elevated expression of cFos protein in the cortex after induction of CSD (Liktor-Busa et al., 2020). The blockade of DAGL$\alpha$ did not induce a significant change in the expression of cFos protein or *cFos* mRNA. Moreover, the application of LEI-106 on brain slices did not initiate the typical increase of light transmittance related to CSD. Together, these data support the role for 2-AG in maintaining the excitatory–inhibitory balance in the cortex; however, the depletion of 2-AG did not induce a CSD event. How reduced 2-AG tone can influence other properties of CSD, such as velocity, duration, spread and area needs to be investigated in the future. A higher concentration of LEI-106 could also have a greater impact on CSD events.

2-AG was previously shown to induce dilatation of peripheral arteries via its action on vascular cannabinoid receptors (Mechoulam et al., 1998). Interestingly, within the cerebral circulation, the effects of 2-AG remain controversial. A study by Hillard et al. (2007) suggests that 2-AG has a mild vasodilatory effect at concentrations up to 1 µM, which is significantly increased in the presence of the metabolic inhibitors URB754 and URB597, suggesting that the modest effect of 2-AG is mostly due to its rapid degradation within the vasculature (Hillard et al., 2007). On the other hand, Shearer et al. (2018) showed that acute application of 2-AG results in the worsening of ischaemic damage in the brain, associated with a more pronounced deficit in cerebral blood flow (Shearer et al., 2018). Our data show that 2-AG, or at least DAGL$\alpha$ activity, has a distinct effect on cerebral microcirculation: while it seemingly does not affect pial artery myogenic constriction, it induces a significant increase in pressure-induced constriction within the parenchymal microcirculation. The mechanisms underlying these differences remain elusive, but it is possible that expression of DAGL$\alpha$ or cannabinoid receptors, of which 2-AG acts as an agonist, differ between branches of the cerebral vascular tree; this, however, remains speculative. Independent of the underlying cause, the observed increase in penetrating arteriole constriction can result in altered haemodynamic regulation. This seems to be the case in our study, as we observed a significant increase in superficial cortical perfusion after acute application of LEI-106. Together, these data suggest that a generalized constriction of penetrating arterioles induced by LEI-106 may lead to an increase in vascular resistance, resulting in increased perfusion of the upstream pial circulation. In addition, the reduced penetrating arteriole diameter will lead to an increase in vascular resistance, thus potentially limiting downstream perfusion and, consequently, shear stress.

Shear stress is a known regulator of BBB permeability, and previous studies showed that physiological shear stress promotes maintenance of BBB integrity by inducing expression of tight junction proteins (Cucullo et al., 2011). On the other hand, increased shear stress was demonstrated to reduce expression of ZO-1 and claudin-5 in primary human brain microvascular endothelial cells, resulting in abnormal morphology of intercellular tight junctions (Garcia-Polite et al., 2017). Shear stress is defined as the force exerted on endothelial cells (static) as a consequence of moving fluid (blood) and can be mathematically expressed as a function of flow, as determined by Pouiselle's law: $\tau = (4\,\mu Q)/(\pi r^3)$, where $Q$ is volumetric flow rate (volume of fluid over time), $\mu$ is fluid viscosity and $r$ is vascular radius. Although Pouiselle's law can only be fully applied to Newtonian fluids assuming rigid cylinders of constant diameter, the implications for physiological haemodynamic regulation hold true and may help explain our findings by connecting vascular radius to shear stress and its effects on BBB integrity. The increased vascular contractility observed after DAGL$\alpha$ inhibition can have two immediate effects: increased shear stress upstream from penetrating arterioles as a consequence of increased volumetric flow, as suggested by our laser speckle contrast imaging findings, which can disrupt tight junctions (Garcia-Polite et al., 2017), as well as reduced shear stress downstream as a consequence of lower volumetric flow. The acute effects of low shear stress on BBB integrity remain poorly understood, but previous studies in the periphery showed that low shear stress can induce expression of the inflammatory mediator nuclear factor-$\kappa$B (NF-$\kappa$B) in endothelial cells (Hu et al., 2018), which can result in BBB disruption (Aveleira et al., 2010). Therefore, acute DAGL$\alpha$ inhibition in penetrating arterioles may lead to a sudden depletion in 2-AG levels, resulting in impaired myogenic regulation and loss of BBB integrity in the cerebral microcirculation by modulating microvascular shear stress.

## Conclusions

Regulation of the BBB is vital for brain health and disease healing for neurological disorders. Data here have identified that endogenous 2-AG plays a role in maintaining tight junctions and BBB function. Thus, the loss of 2-AG tone in disease states can worsen BBB paracellular disruption in a region-dependent manner, as well as in the absence of pathology such as CSD.

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

## Additional information

### Data availability statement

The datasets generated and analysed during the current study are available from the corresponding author upon reasonable request.

### Competing interests

The authors declare they have no competing interests.

### Author contributions

Study design: E.L.B., P.W.P., T.A. and T.M.L. Data collection: E.L.B., A.A.L., S.J.Y., C.B., S.M.P., F.D.P. and S.A.C. Data analysis: E.L.B., A.A.L., S.M.P., P.W.P., F.D.P., T.A. and T.M.L. Writing – original draft: E.L.B. Writing – editing: E.L.B., A.A.L., P.W.P., T.A. and T.M.L. All authors have read and approved the final version of this manuscript and agree to be accountable for all aspects of the work in ensuring that questions related to the accuracy or integrity of any part of the work are appropriately investigated and resolved. All persons designated as authors qualify for authorship, and all those who qualify for authorship are listed.

### Funding

This work was supported by grants from the National Institute of Neurological Disorders and Stroke (R01NS099292 and R01NS126752-01A1, T.M.L.), National Institutes on Aging (R01AG073230, P.W.P.) and Arizona Biomedical Research Commission (ABRC45952, T.M.L.), the Alzheimer's Association (AARGD-21-850835, P.W.P.), with monies from the Department of Pharmacology at the University of Arizona, the M.D.-Ph.D. Program at the University of Arizona, the University of Arizona Comprehensive Pain and Addiction Centre, and the NIDA Centre for Excellence in Addiction Studies (P30DA051355). Research reported in this publication was also supported by the National Cancer Institute of the National Institutes of Health under award number P30CA023074. Authors are solely responsible for the content which does not necessarily represent the official views of the National Institutes of Health, the State of Arizona or the University of Arizona.

### Keywords

2-AG, blood–brain barrier, DAGL$\alpha$, endocannabinoid system, ZO-1

## Supporting information

Additional supporting information can be found online in the Supporting Information section at the end of the HTML view of the article. Supporting information files available:

**Peer Review History**

