## [Peer Review History · The Journal of Physiology]

Inhibition of DAGL α induced blood-brain barrier breach in female Sprague- Dawley rats

Erika Liktor Busa, Aidan A Levine, Sally J Young, Colin Bader, Seph M Palomino, Felipe D Polk, Sarah A Couture, Paulo W. Pires, Trent Anderson, and Tally M Largent-Milnes

DOI: 10.1113/JP287680

Corresponding author(s): Tally Largent-Milnes (tlargent@arizona.edu)

The following individual(s) involved in review of this submission have agreed to reveal their identity: Takeshi Y Hiyama (Referee #1); Sean Williams (Referee #3); Ken D O'Halloran (Referee #4)

Review Timeline:

Submission Date:	11-Sep-2024
Editorial Decision:	16-Oct-2024
Revision Received:	14-Nov-2024
Accepted:	20-Dec-2024

Senior Editor: Harold Schultz

Reviewing Editor: Nikki Jernigan

Transaction Report:

Dear Dr Largent-Milnes,

Re: JP-RP-2024-287680 "Inhibition of DAGL α induced blood-brain barrier breach in female Sprague- Dawley rats" by Erika Liktor Busa, Aidan A Levine, Sally J Young, Colin Bader, Seph M Palomino, Felipe D Polk, Sarah A Couture, Paulo W. Pires, Trent Anderson, and Tally M Largent-Milnes

Thank you for submitting your manuscript to The Journal of Physiology. It has been assessed by a Reviewing Editor and by 4 expert referees and we are pleased to tell you that it is potentially acceptable for publication following satisfactory major revision.

LANGUAGE EDITING AND SUPPORT FOR PUBLICATION: If you would like help with English language editing, or other article preparation support, Wiley Editing Services offers expert help, including English Language Editing, as well as translation, manuscript formatting, and figure formatting at www.wileyauthors.com/eoo/preparation. You can also find resources for Preparing Your Article for general guidance about writing and preparing your manuscript at www.wileyauthors.com/eoo/prepresources.

REVISION CHECKLIST:

We look forward to receiving your revised submission.

Yours sincerely,

Harold Schultz
Senior Editor
The Journal of Physiology

REQUIRED ITEMS

- Author photo and profile. First or joint first authors are asked to provide a short biography (no more than 100 words for one author or 150 words in total for joint first authors) and a portrait photograph. These should be uploaded and clearly labelled together in a Word document with the revised version of the manuscript. See Information for Authors for further details.
- The Journal of Physiology funds authors of provisionally accepted papers to use the premium BioRender site to create high resolution schematic figures. Follow this link and enter your details and the manuscript number to create and download figures. Upload these as the figure files for your revised submission. If you choose not to take up this offer, we require figures to be of similar quality and resolution. If you are opting out of this service to authors, state this in the Comments section on the Detailed Information page of the submission form. The link provided should only be used for the purposes of this submission. Authors will be charged for figures created on this premium BioRender account if they are not related to this manuscript submission.
- Please upload separate high-quality figure files via the submission form.
- You must upload original, uncropped western blot/gel images (including controls) if they are not included in the manuscript. This is to confirm that no inappropriate, unethical or misleading image manipulation has occurred. These should be uploaded as 'Supporting information for review process only'. Please label/highlight the original gels so that we can clearly see which sections/lanes have been used in the manuscript figures. For more information, see: <https://physoc.onlinelibrary.wiley.com/hub/journal-policies#imagmanip>.
- Papers must comply with the Statistics Policy: https://jp.msubmit.net/cgi-bin/main.plex?form_type=display_requirements#statistics.

In summary:

- If n {less than or equal to} 30, all data points must be plotted in the figure in a way that reveals their range and distribution.

A bar graph with data points overlaid, a box and whisker plot or a violin plot (preferably with data points included) are acceptable formats.

- If $n > 30$, then the entire raw dataset must be made available either as supporting information, or hosted on a not-for-profit repository, e.g. FigShare, with access details provided in the manuscript.

- 'n' clearly defined (e.g. x cells from y slices in z animals) in the Methods. Authors should be mindful of pseudoreplication.

- All relevant 'n' values must be clearly stated in the main text, figures and tables.

- The most appropriate summary statistic (e.g. mean or median and standard deviation) must be used. Standard Error of the Mean (SEM) alone is not permitted.

- Exact p values must be stated. Authors must not use 'greater than' or 'less than'. Exact p values must be stated to three significant figures even when 'no statistical significance' is claimed.

- Please include an Abstract Figure file, as well as the Figure Legend text within the main article file. The Abstract Figure is a piece of artwork designed to give readers an immediate understanding of the research and should summarise the main conclusions. If possible, the image should be easily 'readable' from left to right or top to bottom. It should show the physiological relevance of the manuscript so readers can assess the importance and content of its findings. Abstract Figures should not merely recapitulate other figures in the manuscript. Please try to keep the diagram as simple as possible and without superfluous information that may distract from the main conclusion(s). Abstract Figures must be provided by authors no later than the revised manuscript stage and should be uploaded as a separate file during online submission labelled as File Type 'Abstract Figure'. Please also ensure that you include the figure legend in the main article file. All Abstract Figures should be created using BioRender. Authors should use The Journal's premium BioRender account to export high-resolution images. Details on how to use and access the premium account are included as part of this email.

Reviewing Editor's comments:

This manuscript addresses an important gap, as few studies explore the molecular mechanisms of endocannabinoid regulation of the blood-brain barrier (BBB) function in vivo. Therefore, this study has the potential to offer valuable new insights and could be highly impactful, with implications for both basic biology and research related to cannabis use. However, while the research question is relevant to J. Physiology and the study is promising, the consistency of the data and the overall completeness are lacking. With major revisions, along the lines of reviewer suggestions, to clarify the data and resolve experimental inconsistencies, this could become a significant contribution to the field.

Senior Editor:

Comments to ensure the paper complies with the Statistics Policy:

Please include the actual p values in the figures. Do not use symbols or NS.

Comments to the Authors:

Thank you for submission of your research article to the Journal of Physiology for consideration. The article has been reviewed by experts in the field and found to be potentially acceptable for publication pending a major revision to address all of the concerns raised. Please address all comments from the external referees and reviewing editor as well as addressing the list of requirements or publication listed in this letter and in the Journal's requirements for Rigour and Reproducibility (see link below). A revised manuscript must follow the formatting requirements for the Journal as described in the Instructions to the Authors. Please provide a copy of the revised manuscript with changes highlighted and so named.

link to R&R

<https://physoc.onlinelibrary.wiley.com/pb-assets/hub-assets/physoc/documents/TJP-Rigour-and-Reproducibility-Requirements-1724673661727.pdf>

Animal ethics

Please review the Journal's policy on reporting animal experiments outlined in the animal ethics check list:

<https://physoc.onlinelibrary.wiley.com/hub/animal-experiments>

1. The Methods should begin with subheading "Ethical Approval". In addition to information required as stated in the check list, this section must include a statement that the investigators understand the ethical principles under which the journal operates and that their work complies with the animal ethics checklist as outlined in the Journal policy.

2. All surgical procedures must describe anaesthetic protocols in detail (premedication, anaesthetic(s) used, dose, route, supplementary dosage). Describe how the depth of anaesthesia was determined and maintained through the surgery and frequency of determination. For recovery surgery, describe appropriate post-surgical care and analgesia and steps taken to ensure that pain or distress, whether physical or psychological, was minimized. For terminal surgery with experimental interventions, describe how the depth of anaesthesia was determined and maintained through the experiment and frequency of determination in relation to experimental procedures. This is particularly important when the dose of supplemental anaesthesia was reduced. Describe the approximate duration of the experiment and the end point. Describe how animals were euthanized at the end point (anaesthetic(s) used, dose, route), and how death was ascertained. Similarly, describe how animals were euthanized and confirmed dead for collection of tissue outside of a terminal experiment.

Western blots

Several of the western blot examples in Figure 2 do not best represent the summary data in the graphs. An example is the tubulin loading control for ZO-1 in panel C is markedly different between Veh and LEI-106, but not ZO-1. This is not consistent with the summary data in the graph. Please ensure that all western blots in the figures provide sufficient space above and below the blot to encompass the total protein.

Statistics

Please include the actual p values in the figures. Do not use symbols or NS. This will help to streamline the figure legends, which are rather lengthy.

Referee #1:

This paper discusses the molecular mechanisms of BBB disruption due to depletion of 2-AG in the BBB. The authors demonstrated that intraperitoneal administration of the DAGL α inhibitor, LEI-106, causes BBB disruption in the cerebral cortex and PAG. In the cerebral cortex, LEI-106 directly acts to inhibit the production of the endocannabinoid 2-AG via DAGL α , resulting in the loss of 2-AG's protective effect on the BBB. They showed that the decrease in ZO-1 protein levels is the cause of BBB disruption. However, as there was no difference at the mRNA level, they argue that the decrease occurred not at the transcriptional level but at the translational level. Additionally, although there was an increase in the immediate early gene transcription factor NPAS4, there was no increase in cFOS. On the other hand, in the PAG, 2-AG levels were unchanged, and the authors claimed that LEI-106 did not reach the PAG directly. In the PAG, there was no decrease in ZO-1 protein levels or increase in NPAS4. They also demonstrated that LEI-106 increased 2-AG levels in the serum, but the mechanism and its relationship with 2-AG levels in the brain remain unclear. Finally, they dissected the penetration artery and pial artery and investigated the myogenic tone caused by LEI-106, showing that it only enhanced the myogenic tone of the penetration artery. They also showed an increase in cerebral surface blood flow using optical measurements.

This paper provides intriguing data and has publication value; however, the following points need to be addressed before acceptance.

Major points:

#1. Do the authors assume that LEI-106 can pass through the BBB, while 2-AG cannot? If so, they need to explain why the increased LEI-106 in the cerebrospinal fluid did not reach the PAG. If there is any information on the extent to which LEI-106 reaches different areas of the brain, it should be presented as well.

#2. The authors stated that 2-AG concentration increased in the serum as a compensatory mechanism. Please discuss about the source of 2-AG in the periphery independent of DAGL α . Whether the increase in serum 2-AG influenced the 2-AG levels in the cerebrospinal fluid would be a crucial piece of data central to this paper. The authors need to explain, through literature or experiments, how peripheral 2-AG might influence cerebrospinal fluid 2-AG levels. In line 695, the authors stated that 2-AG is rapidly degraded within blood vessels (Hillard et al., 2007), but what is the relationship with the increase in serum 2-AG?

#3. There was no decrease in ZO-1 expression in the PAG. The authors should explain how the molecular mechanisms of BBB disruption differ between cortex and PAG.

#4. It is not clearly stated how the increase in tone in penetrating arterioles relates to the increased cerebral surface blood flow. Do the authors suggest that as a result of the increased tone in the penetrating arterioles, more blood is distributed to the pial arteries, leading to an increase in cerebral surface blood flow? Please clarify.

Minor points:

The legends for Figure 6F is missing, and the description of the arrows is also absent.

Referee #2:

Liktor-Busa et al. present a study of the effects of LEI-106 on BBB sucrose permeability, ZO-1 and NPAS4 expression and vascular reactivity in female rats.

Overall it is an interesting study, and a research question that I believe would be of interest to the readers of J. Phys. However, in my view, some of the presented data are not convincing. Additionally the results of different experiments are not consistent with one another but this is not discussed in any detail. I have the following comments, suggestions, and questions for the authors:

1. The authors use only female rats, but do not explain this decision anywhere in the manuscript. Why were male rats not also studied? What was the motivation for the choice?

2. In some experiments isoflurane was used as anaesthetic, whereas in other K/X was used. Different anaesthetics have profoundly different effects on the activity of the glymphatic system (DOI: 10.1126/sciadv.aav5447), which could have knock-on effects on measured BBB permeability. Could the authors explain why different anaesthetics are used for different experiments? They could also discuss potential differential effects of different anaesthetics.

3. The changes in sucrose permeability reported in figure 1B are quite modest. Are the authors convinced that they are physiologically relevant? It would be very helpful to include a positive control BBB-disrupting agent so that the reader can assess the relevance of this relatively small effect size. It would also be helpful to explain the rationale for choosing this method over a more standard measure such as Evans Blue extravasation.

4. The authors propose that the mechanism of action by which LEI-106 increases BBB sucrose permeability is the decrease in ZO-1 protein abundance in the cortex, but not in the PAG. In my view is it unlikely that the same drug has the same physiological effect in different brain regions but by completely different mechanisms. However, the ZO-1 abundance data is not convincing - in the representative cortex ZO-1 blot in Figure 2, it looks like ZO-1 is completely absent in one rat, but not really changed in the other two. Looking at the individual data points in the bar chart, it looks like there might be one low ZO-1 outlier (was this checked?) that is skewing the results.

5. The authors report that LEI-106 had no effect on 2-AG abundance in the PAG (figure 3), despite the fact that it changed BBB sucrose permeability (figure 1). To me, this strongly suggests an off-target effect of LEI-106 on BBB sucrose permeability, possibly independent of endocannabinoid signalling, but this is not discussed.

5a. LEI-106 has also been reported to inhibit the 2-AG hydrolysing enzyme ABHD6 (e.g. doi: 10.1194/jlr.D056390), but this is not discussed. What are the implications for this on the authors' results? Additionally, the authors propose that the reported increase in serum 2-AG is due to a "compensatory mechanism" (line 645). Could an alternative explanation be inhibition of 2-AG hydrolysis by LEI-106?

5b. How well-validated is this inhibitor generally? Has anyone investigated what other proteins LEI-106 can bind/inhibit besides DAGLa/ABHD6 (by e.g. PISA)? It would be helpful if off-target effects could be discussed. Addition of a second DAGLa inhibitor (ideally based on a different lead structure) would be very helpful to clarify the results here.

6a. The NPAS4 data is not clear to me. Firstly, the increase in NPAS4 protein abundance is quite modest. Does this small change in protein abundance actually lead to any changes in transcription of NPAS4 target genes? It would be helpful to include qPCR data on a few NPAS4 target genes.

6b. Secondly, as an immediate early gene, NPAS4 is thought to be regulated at the level of transcription (i.e. NPAS4 transcript increases upon stimulation). However, the authors show no change in NPAS4 gene expression. Indeed, in the text the authors say that the NPAS4 qPCR was done to "confirm the results obtained from Western immunoblotting" (line 465). However, the qPCR results do not confirm the results of the western blotting, but this is not discussed. Do the authors propose that this change in NPAS4 protein abundance in the absence of transcriptional change is a novel NPAS4 regulatory mechanism?

Overall, whilst this is an interesting research question that is clearly in remit for J. Phys., in my view the manuscript in its present form is not suitable for publication in this journal.

Referee #3:

The statistical analyses are broadly appropriate but would benefit from additional clarity and detail. First, it would be important to specify whether assumptions of normality and sphericity were checked prior to applying parametric tests, such as t-tests and repeated measures ANOVA. For normality, the Shapiro-Wilk test should be mentioned, while for repeated measures ANOVA, checking for sphericity using the Greenhouse-Geisser epsilon should be detailed (see Laerd Statistics on Sphericity). If the Greenhouse-Geisser epsilon (ϵ) is less than 0.75, the Greenhouse-Geisser correction should be used for p-values, while the Huynh-Feldt correction may be applied if the estimated epsilon is greater than 0.75.

The results/figures refer to 'mixed-model two-way ANOVA with a Šídák correction for multiple comparisons,' but this is not described in the methods section. Adding an explanation of this analysis would help ensure consistency between the methods and results sections.

While the inclusion of a sample size justification is commendable, it requires more detail for replication and transparency. Specifically, what effect size was used in the GPower calculation, and what was the rationale for its selection? The mention of a "20% difference" lacks clarity, as this is not a direct input in GPower, and there is no information about which variable this relates to or why a 20% difference is meaningful in the context of the study.

Finally, three different versions of GraphPad Prism (7.0, 8.3.1, and 10) are listed. If all of these were necessary, it would be helpful to clarify which version was used for each type of analysis to avoid confusion and ensure reproducibility.

Referee #4:

Thank you for submitting your manuscript to The Journal of Physiology. Some important additional details pertaining to ethics and animal welfare are required.

1. You must start the Methods section with the subheading "Ethical approval". Add the institutional approval code/number for the study.
2. Line 120: Add the route of administration of anaesthetic. How was depth of anaesthesia determined? Add details.
3. Line 122: State here that animals were killed by exsanguination by perfusion. Since the perfusion persisted for 15mins, which at the flow rate used, was required to fully clear blood volume, please state how a sufficient depth of anaesthesia was determined before and during the perfusion.
4. Line 140: As above, although it may appear redundant, please add details on how depth of anaesthesia was determined. The fate of all animals must be fully and clearly accounted for. Give the route of administration of the anaesthetic.
5. Line 214: State 'killed' or 'euthanised' not sacrificed. State decapitation for clarity.

Line 253: How was depth of anaesthesia confirmed. Add details.

Line 265: What assessments were made to confirm an adequate depth of anaesthesia was maintained?

END OF COMMENTS

Response to Referees

The authors would like to thank the senior editor and the referees for their constructive comments and suggestions. The authors' answers for each comment (in Italic, highlighted with grey) can be found below. The changes and edits in the manuscript are highlighted in yellow.

Senior Editor:

Comments to ensure the paper complies with the Statistics Policy:

Please include the actual p values in the figures. Do not use symbols or NS.

As requested by the senior editor, the actual p values have been added in the revised figures.

Comments to the Authors:

Thank you for submission of your research article to the Journal of Physiology for consideration. The article has been reviewed by experts in the field and found to be potentially acceptable for publication pending a major revision to address all of the concerns raised. Please address all comments from the external referees and reviewing editor as well as addressing the list of requirements or publication listed in this letter and in the Journal's requirements for Rigour and Reproducibility (see link below). A revised manuscript must follow the formatting requirements for the Journal as described in the Instructions to the Authors. Please provide a copy of the revised manuscript with changes highlighted and so named.

link to R&R

<https://physoc.onlinelibrary.wiley.com/pb-assets/hub-assets/physoc/documents/TJP-Rigour-and-Reproducibility-Requirements-1724673661727.pdf>

Animal ethics

Please review the Journal's policy on reporting animal experiments outlined in the animal ethics check list:

<https://physoc.onlinelibrary.wiley.com/hub/animal-experiments>

1. The Methods should begin with subheading "Ethical Approval". In addition to information required as stated in the check list, this section must include a statement that the investigators understand the ethical principles under which the journal operates and that their work complies with the animal ethics checklist as outlined in the Journal policy.

As requested, the section of "Ethical Approval" has been added in the revised version of the manuscript.

2. All surgical procedures must describe anaesthetic protocols in detail (premedication, anaesthetic(s) used, dose, route, supplementary dosage). Describe how the depth of anaesthesia was determined and maintained through the surgery and frequency of determination. For recovery surgery, describe appropriate post-surgical care and analgesia and steps taken to ensure that pain or distress, whether physical or psychological, was minimized. For terminal surgery with experimental interventions, describe how the depth of anaesthesia was determined and maintained through the experiment and frequency of determination in relation to experimental procedures. This is particularly important when the dose of supplemental anesthesia was reduced. Describe the approximate duration of the experiment and the end point. Describe how animals were euthanized at the end point (anaesthetic(s) used, dose, route), and how death was ascertained. Similarly, describe how animals were euthanized and confirmed dead for collection of tissue outside of a terminal experiment.

All required details have been added in the revised version of the manuscript (see more details in the answer for Referee#4).

Western blots

Several of the western blot examples in Figure 2 do not best represent the summary data in the graphs. An example is the tubulin loading control for ZO-1 in panel C is markedly different between Veh and LEI-106, but not ZO-1. This is not consistent with the summary data in the graph. Please ensure that all western blots in the figures

provide sufficient space above and below the blot to encompass the total protein.

Better representative blots have been provided. Sufficient space below and above the blots have been added.

Statistics

Please include the actual p values in the figures. Do not use symbols or NS. This will help to streamline the figure legends, which are rather lengthy.

As requested by the senior editor, the actual p values have been added in the figures.

Referee #1:

This paper discusses the molecular mechanisms of BBB disruption due to depletion of 2-AG in the BBB. The authors demonstrated that intraperitoneal administration of the DAGL α inhibitor, LEI-106, causes BBB disruption in the cerebral cortex and PAG. In the cerebral cortex, LEI-106 directly acts to inhibit the production of the endocannabinoid 2-AG via DAGL α , resulting in the loss of 2-AG's protective effect on the BBB. They showed that the decrease in ZO-1 protein levels is the cause of BBB disruption. However, as there was no difference at the mRNA level, they argue that the decrease occurred not at the transcriptional level but at the translational level. Additionally, although there was an increase in the immediate early gene transcription factor NPAS4, there was no increase in cFOS. On the other hand, in the PAG, 2-AG levels were unchanged, and the authors claimed that LEI-106 did not reach the PAG directly. In the PAG, there was no decrease in ZO-1 protein levels or increase in NPAS4. They also demonstrated that LEI-106 increased 2-AG levels in the serum, but the mechanism and its relationship with 2-AG levels in the brain remain unclear. Finally, they dissected the penetration artery and pial artery and investigated the myogenic tone caused by LEI-106, showing that it only enhanced the myogenic tone of the penetration artery. They also showed an increase in cerebral surface blood flow using optical measurements.

This paper provides intriguing data and has publication value; however, the following points need to be addressed before acceptance.

Major points:

#1. Do the authors assume that LEI-106 can pass through the BBB, while 2-AG cannot? If so, they need to explain why the increased LEI-106 in the cerebrospinal fluid did not reach the PAG. If there is any information on the extent to which LEI-106 reaches different areas of the brain, it should be presented as well.

The authors do not assume that LEI-106 can pass through BBB, while 2-AG cannot. We did not claim either that the increased LEI-106 in the cerebrospinal fluid did not reach the PAG. We clarified this point in the text. The unchanged 2-AG level in PAG after administration of LEI-106 might be an off-target effect – see more about this and other possibilities under the next comment. LEI-106 was characterized using mouse brain membranes (doi: 10.1021/jm500681z.), but little is known about the brain availability of LEI-106. However, it is known that the distribution of DAGL α is brain-region specific

(doi: 10.1016/j.neuroscience.2011.06.062), therefore it is possible that the effect of LEI-106 might be different in distinct brain areas. Details about this concept have been added to the discussion section.

#2. The authors stated that 2-AG concentration increased in the serum as a compensatory mechanism. Please discuss about the source of 2-AG in the periphery independent of DAGL α . Whether the increase in serum 2-AG influenced the 2-AG levels in the cerebrospinal fluid would be a crucial piece of data central to this paper. The authors need to explain, through literature or experiments, how peripheral 2-AG might influence cerebrospinal fluid 2-AG levels. In line 695, the authors stated that 2-AG is rapidly degraded within blood vessels (Hillard et al., 2007), but what is the relationship with the increase in serum 2-AG?

The following discussion has been added to the revised version:

“The circulating endocannabinoids can originate from multiple organs, including brain, muscle, adipose tissue, and circulating blood cells (Hillard, 2018). The β isoform of DAGL is the main enzyme responsible for the production of 2-AG in non-neural tissue (Reisenberg et al., 2012), therefore the serum level of 2-AG might be independent from the activity of DAGL α . Results published by Shonesy and co-workers support this hypothesis, since normal peripheral 2-AG level was found in DAGL α KO mice, however the brain 2-AG levels were robustly reduced (Shonesy et al., 2014).

There are very few papers available discussing the correlation between serum and cerebrospinal level of endocannabinoids. A weak, but positive correlation was found between serum and cerebrospinal fluid levels of 2-AG in multiple sclerosis patients (Meier et al., 2024). The level of 2-AG in cerebrospinal fluid was not assessed in this study, but the elevated serum 2-AG level might induce increased 2-AG concentration in the cerebrospinal fluid, which may have an impact on the 2-AG level in the PAG as the cerebral aqueduct is surrounded by the PAG. This is just one possible explanation for the unchanged 2-AG level in the PAG after a higher dose of DAGL α blockade. Other explanations include the possible off-target effects of LEI-106 or the different distribution of DAGL α throughout the central nervous system. It is known that LEI-106 has off-targets, like ABHD6, which is an enzyme partially responsible for the degradation of 2-AG. The IC₅₀ value of LEI-106 for DAGL α (18 nM) is much lower than its value for ABHD6 (0.8 μ M) (Janssen et al., 2014), but the higher dose of LEI-106 might mitigate this difference. Our former data indicate that the administration of LEI-106 has region-specific impact on the 2-AG levels, no reduction in the level of 2-AG was found in trigeminal ganglion or trigeminal nucleus caudalis after systemic LEI-106 injection (Levine 2021), suggesting different distribution of DAGL α within the central nervous system. This hypothesis is supported by literature evidence as well (Suárez et al.,

2011). Additional studies beyond the scope of the current work are warranted to elucidate these mechanisms.

#3. There was no decrease in ZO-1 expression in the PAG. The authors should explain how the molecular mechanisms of BBB disruption differ between cortex and PAG.

Since the level of 2-AG in PAG was not significantly reduced after LEI-106 administration and no change in ZO-1 expression was found, the BBB opening in PAG might be 2-AG-independent or it might be the result of off-target effects. Discussion section has been expanded to highlight these possibilities.

#4. It is not clearly stated how the increase in tone in penetrating arterioles relates to the increased cerebral surface blood flow. Do the authors suggest that as a result of the increased tone in the penetrating arterioles, more blood is distributed to the pial arteries, leading to an increase in cerebral surface blood flow? Please clarify.

Although we did not strongly argued this point in the original submission, it is likely that the increase in cerebral perfusion observed by laser speckle contrast imaging (Figure 6) is a consequence of widespread contraction of penetrating arterioles. Our pressure myography experiment corroborate this possibility, as we observed that myogenic reactivity of surface pial arteries was unchanged by LEI-106, whereas pressure-induced contractility of penetrating arterioles was significantly higher at physiologically relevant levels of intraluminal pressure. We included the following sentence in the Discussion clarifying this point:

Independent of the underlying cause, the observed increase in penetrating arteriole constriction can result in altered hemodynamic regulation. This seem to be the case in our study, as we observed a significant increase in superficial cortical perfusion after acute application of LEI-106. Together, these data suggest that a generalized constriction of penetrating arterioles induced by LEI-106 may lead to an increase in vascular resistance, resulting in increased perfusion of the upstream pial circulation. In addition, the reduced penetrating arteriole diameter will lead to an increase in vascular resistance, thus potentially limiting downstream perfusion and, consequently, shear stress.

Minor points:

The legends for Figure 6F is missing, and the description of the arrows is also absent.

We apologize for our mistake in the original submission and included the legend for Figure 6F and a description of the arrow in the image (see below).

(E-G) LEI-106 (10 μ M) caused an acute increase in superficial cortical perfusion when applied on top of the thinned-skull cranial window (left hemisphere in E), blue trace in F) when compared to simultaneous application of vehicle (0.02% DMSO in aCSF, right hemisphere in E), gray trace in F). The traces in F show that there was no cross-contamination between the different hemispheres (the sagittal suture was not thinned, forming a barrier separating right and left hemispheres). The arrow in F indicate when LEI-106 was applied to the thinned-skull cranial window. The significant increase in perfusion can be seen in the summary graph and it was assessed by unpaired t-test (G). All data are shown as the % of baseline \pm SD (n=7 female Sprague-Dawley rats).

Referee #2:

Liktor-Busa et al. present a study of the effects of LEI-106 on BBB sucrose permeability, ZO-1 and NPSA4 expression and vascular reactivity in female rats.

Overall it is an interesting study, and a research question that I believe would be of interest to the readers of J. Phys. However, in my view, some of the presented data are not convincing. Additionally the results of different experiments are not consistent with one another but this is not discussed in any detail. I have the following comments, suggestions, and questions for the authors:

1. The authors use only female rats, but do not explain this decision anywhere in the manuscript. Why were male rats not also studied? What was the motivation for the choice?

Headache disorders, including migraine is associated with disruption of blood-brain barrier permeability. It is known that 70% of migraine patients are woman. The higher prevalence of migraine disorder in woman population was one reason of the inclusion of female animals. In our former work (Levine et al, 2021, doi: 10.3389/fphar.2020.615028), we demonstrated that female SD rats expressed significantly higher periorbital sensitivity after DAGL α inhibition compared to male animals, suggesting sexual dimorphism and providing rational to use of female animals in this study. We highlighted this in the revised version.

2. In some experiments isoflurane was used as anaesthetic, whereas in other K/X was used. Different anaesthetics have profoundly different effects on the activity of the lymphatic system (DOI: 10.1126/sciadv.aav5447), which could have knock-on effects on measured BBB permeability. Could the authors explain why different anaesthetics are used for different experiments? They could also discuss potential differential effects of different anaesthetics.

The administration of ketamine/xylazine as anesthetic is required for the in-situ brain perfusion experiments, since radioactive materials involved in this experiment can cause potential RAM contamination of the isoflurane vaporizer. We do agree with the reviewer that different anesthetic can have different effect of the lymphatic system, further on the BBB permeability, but results after the administration of LEI-106 was always compared to vehicle control in which group the animals were subjected to the same anesthetic method. How different anesthetic models can influence the effect of LEI-106 on BBB permeability should be investigated in future projects.

3. The changes in sucrose permeability reported in figure 1B are quite modest. Are the authors convinced that they are physiologically relevant? It would be very helpful to include a positive control BBB-disrupting agent so that the reader can assess the relevance of this relatively small effect size. It would also be helpful to explain the rationale for choosing this method over a more standard measure such as Evans Blue extravasation.

In our former work (Cottier et al., 2018, doi: 10.1523/ENEURO.0116-18.2018), about 50% increase of C¹⁴-sucrose uptake in cortex was observed after induction of cortical spreading depression (cortical injection of KCl) compared to control. In the current model, the increase of C¹⁴-sucrose uptake was lower (about 20-30%), but along with the observed changes in the ZO-1 expression and the former behavior data (Levine et al, 2021, doi: 10.3389/fphar.2020.615028), the physiological relevance is suggested. Moreover, the light transmittance experiments showed the absence of cortical spreading depression after LEI-106 application, therefore the degree of LEI-106 impact on the BBB is quite expected.

Evans Blue and sucrose have different molecular weights and molecular sizes. Evans Blue can bind to serum albumin; therefore, Evans Blue can be utilized to detect BBB leakage of molecules with higher molecular weight. Its color can provide easy visualization of the altered brain area, but quantification of the BBB impact would be challenging with Evans-Blue. The usage of radioactive sucrose in in-situ brain perfusion experiment is suitable and a standard method to reveal the opening of blood brain barrier for small molecules (doi: 10.1186/s12987-020-00230-3). Sucrose does not have any known transporter in the neurovascular unit, and it can be utilized for the quantitative determination of the changes in blood-brain permeability.

4. The authors propose that the mechanism of action by which LEI-106 increases BBB sucrose permeability is the decrease in ZO-1 protein abundance in the cortex, but not in the PAG. In my view is it unlikely that the same drug has the same physiological effect in different brain regions but by completely different mechanisms. However, the ZO-1 abundance data is not convincing - in the representative cortex ZO-1 blot in Figure 2, it looks like ZO-1 is completely absent in one rat, but not really changed in the other two. Looking at the individual data points in the bar chart, it looks like there might be one low ZO-1 outlier (was this checked?) that is skewing the results.

Thank you for this remark. All data set, including this one, was checked for outliers. The data point mentioned above was not identified as outlier.

5. The authors report that LEI-106 had no effect on 2-AG abundance in the PAG (figure 3), despite the fact that it changed BBB sucrose permeability (figure 1). To me, this strongly suggests an off-target effect of LEI-106 on BBB sucrose permeability, possibly independent of endocannabinoid signalling, but this is not discussed.

The following discussion has been added to the revised version:

“The level of 2-AG in PAG was not significantly reduced after LEI-106 administration and no change in ZO-1 expression was found. These results suggest that the BBB opening in PAG after LEI-106 administration might be independent from endocannabinoid signaling or caused by off-target effects.”

5a. LEI-106 has also been reported to inhibit the 2-AG hydrolysing enzyme ABHD6 (e.g. doi: 10.1194/jlr.D056390), but this is not discussed. What are the implications for this on the authors' results? Additionally, the authors propose that the reported increase in serum 2-AG is due to a "compensatory mechanism" (line 645). Could an alternative explanation be inhibition of 2-AG hydrolysis by LEI-106?

The following discussion has been added to the revised version:

“The circulating endocannabinoids can originate from multiple organs, including brain, muscle, adipose tissue, and circulating blood cells (Hillard, 2018). The β isoform of DAGL is the main enzyme responsible for the production of 2-AG in non-neural tissue (Reisenberg et al., 2012), therefore the serum level of 2-AG might be independent from the activity of DAGL α . Results published by Shonesy and co-workers support this hypothesis, since normal peripheral 2-AG level was found in DAGL α KO mice, however the brain 2-AG levels were robustly reduced (Shonesy et al., 2014).

There are very few papers available discussing the correlation between serum and cerebrospinal level of endocannabinoids. A weak, but positive correlation was found between serum and cerebrospinal fluid levels of 2-AG in multiple sclerosis patients (Meier et al., 2024). The level of 2-AG in cerebrospinal fluid was not assessed in this study, but the elevated serum 2-AG level might induce increased 2-AG concentration in the cerebrospinal fluid, which may have an impact on the 2-AG level in the PAG as the cerebral aqueduct is surrounded by the PAG. This is just one possible explanation for the unchanged 2-AG level in the PAG after a higher dose of DAGL α blockade. Other explanations include the possible off-target effects of LEI-106 or the different distribution of DAGL α throughout the central nervous system. It is known that LEI-106 has off-

targets, like ABHD6, which is an enzyme partially responsible for the degradation of 2-AG. The IC50 value of LEI-106 for DAGL α (18 nM) is much lower than its value for ABHD6 (0.8 μ M) (Janssen et al., 2014), but the higher dose of LEI-106 might mitigate this difference. Our former data indicate that the administration of LEI-106 has region-specific impact on the 2-AG levels, no reduction in the level of 2-AG was found in trigeminal ganglion or trigeminal nucleus caudalis after LEI-106 injection (Levine 2021), suggesting different distribution of DAGL α within the central nervous system. This hypothesis is supported by literature evidence as well (Suárez et al., 2011). Additional studies beyond the scope of the current work are warranted to elucidate these mechanisms.

5b. How well-validated is this inhibitor generally? Has anyone investigated what other proteins LEI-106 can bind/inhibit besides DAGL α /ABHD6 (by e.g. PISA)? It would be helpful if off-target effects could be discussed. Addition of a second DAGL α inhibitor (ideally based on a different lead structure) would be very helpful to clarify the results here.

ABHD6 is the only one known off-target of LEI-106 (doi: 10.1021/jm500681z). In the revised version, the possibility of the off-target effect has been discussed. The authors do agree with the reviewer that additional experiments with a second DAGL α inhibitor would be helpful to clarify results, but it is beyond the scope of the current manuscript.

6a. The NPAS4 data is not clear to me. Firstly, the increase in NPAS4 protein abundance is quite modest. Does this small change in protein abundance actually lead to any changes in transcription of NPAS4 target genes? It would be helpful to include qPCR data on a few NPAS4 target genes.

Nptx2, Plk2 and Bdnf are the well-known target genes of NPAS4, however a recent paper (doi: 10.1038/s41586-023-05711-7) identified 1,766 new targets of NPAS4 in the hippocampus. Moreover, the target genes of NPAS4 can be cell-type-specific, therefore the identification of NPAS4 target genes in the cortical areas involved in LEI-106 effect or in BBB regulation is beyond the scope of the current manuscript.

6b. Secondly, as an immediate early gene, NPAS4 is thought to be regulated at the level of transcription (i.e. NPAS4 transcript increases upon stimulation). However, the authors show no change in NPAS4 gene expression. Indeed, in the text the authors say that the NPAS4 qPCR was done to "confirm the results obtained from Western immunoblotting" (line 465). However, the qPCR results do not confirm the results of the western blotting,

but this is not discussed. Do the authors propose that this change in NPAS4 protein abundance in the absence of transcriptional change is a novel NPAS4 regulatory mechanism?

The authors do not propose a novel NPAS4 regulatory mechanism. The qPCR results do not confirm the results of the western blotting, which has been clarified in the revised version. Only one time-point was assessed after LEI-106 administration, we do not have any data about the instant effect of DAG1a blockade. There is a possibility that transcriptional changes might occur at earlier time-points. Discussion has been expanded to address this comment.

Overall, whilst this is an interesting research question that is clearly in remit for J. Phys., in my view the manuscript in its present form is not suitable for publication in this journal.

Referee #3:

The statistical analyses are broadly appropriate but would benefit from additional clarity and detail. First, it would be important to specify whether assumptions of normality and sphericity were checked prior to applying parametric tests, such as t-tests and repeated measures ANOVA. For normality, the Shapiro-Wilk test should be mentioned, while for repeated measures ANOVA, checking for sphericity using the Greenhouse-Geisser epsilon should be detailed (see Laerd Statistics on Sphericity). If the Greenhouse-Geisser epsilon (ϵ) is less than 0.75, the Greenhouse-Geisser correction should be used for p-values, while

The results/figures refer to 'mixed-model two-way ANOVA with a Šídák correction for multiple comparisons,' but this is not described in the methods section. Adding an explanation of this analysis would help ensure consistency between the methods and results sections.

Thank you for this comment, assumptions of normality and sphericity were checked prior to applying parametric measures. Šídák correction has been added into the revised version of the manuscript.

While the inclusion of a sample size justification is commendable, it requires more detail for replication and transparency. Specifically, what effect size was used in the GPower calculation, and what was the rationale for its selection? The mention of a "20% difference" lacks clarity, as this is not a direct input in GPower, and there is no information about which variable this relates to or why a 20% difference is meaningful in the context of the study.

This phrasing has been changed to reflect the effect size (ρ) = 0.2 as calculated in GPower3.1.9.7

Finally, three different versions of GraphPad Prism (7.0, 8.3.1, and 10) are listed. If all of these were necessary, it would be helpful to clarify which version was used for each type of analysis to avoid confusion and ensure reproducibility.

The authors would like to apologize. Three different versions of GraphPad were mistakenly stated in the original version. Only GraphPad 9.5 was utilized in the entire study. It has been corrected in the revised version.

Referee #4:

Thank you for submitting your manuscript to The Journal of Physiology. Some important additional details pertaining to ethics and animal welfare are required.

1. You must start the Methods section with the subheading "Ethical approval". Add the institutional approval code/number for the study.

As requested, the section of "Ethical Approval" has been added in the revised version of the manuscript.

2. Line 120: Add the route of administration of anaesthetic. How was depth of anaesthesia determined? Add details.

Details were added in the revised version of the manuscript.

3. Line 122: State here that animals were killed by exsanguination by perfusion. Since the perfusion persisted for 15mins, which at the flow rate used, was required to fully clear blood volume, please state how a sufficient depth of anaesthesia was determined before and during the perfusion.

The animals were killed by decapitation as stated in line 131 in the original version of the manuscript. Details about the monitoring the depth of anaesthesia have been added in the revised version of the manuscript.

4. Line 140: As above, although it may appear redundant, please add details on how depth of anaesthesia was determined. The fate of all animals must be fully and clearly accounted for. Give the route of administration of the anaesthetic.

Details about the monitoring the depth of anaesthesia and the route of anaesthetic have been added in the revised version of the manuscript. As stated in line 142 in the original version of the manuscript, the animals were killed by decapitation.

5. Line 214: State 'killed' or 'euthanised' not sacrificed. State decapitation for clarity.

All suggestions have been added in the revised version of the manuscript.

Line 253: How was depth of anaesthesia confirmed. Add details.

Details have been added in the revised version of the manuscript.

Line 265: What assessments were made to confirm an adequate depth of anaesthesia was maintained?

Details in the beginning of this section have been added in the revised version.

Dear Dr Largent-Milnes,

Re: JP-RP-2024-287680R1 "Inhibition of DAGL α induced blood-brain barrier breach in female Sprague- Dawley rats" by Erika Liktor Busa, Aidan A Levine, Sally J Young, Colin Bader, Seph M Palomino, Felipe D Polk, Sarah A Couture, Paulo W. Pires, Trent Anderson, and Tally M Largent-Milnes

Thank you for being so willing and responsive to our recent queries over the image checking process.

We are pleased to tell you that your paper has been accepted for publication in The Journal of Physiology.

Yours sincerely,

Harold Schultz
Senior Editor
The Journal of Physiology

If you would like to receive our 'Research Roundup', a monthly newsletter highlighting the cutting-edge research published in The Physiological Society's family of journals (The Journal of Physiology, Experimental Physiology, Physiological Reports, The Journal of Nutritional Physiology and The Journal of Precision Medicine: Health and Disease), please click this link, fill in your name and email address and select 'Research Roundup':
<https://www.physoc.org/journals-and-media/membernews>

- You can help your research get the attention it deserves! Check out Wiley's free Promotion Guide for best-practice recommendations for promoting your work at: www.wileyauthors.com/eeo/guide. You can learn more about Wiley Editing Services which offers professional video, design, and writing services to create shareable video abstracts, infographics, conference posters, lay summaries, and research news stories for your research at: www.wileyauthors.com/eeo/promotion.

EDITOR COMMENTS

Reviewing Editor:

The authors have adequately discussed concerns.

Senior Editor:

The editors thank the authors for these final adjustments to the manuscript. The article is now accepted for publication. Congratulations for an interesting study. Please consider the Journal of Physiology for your future studies.

REFeree COMMENTS

Referee #1:

I agree to publication.

Referee #2:

In the response letter, where additional experiments were suggested the authors generally agree with the points raised but say any further work is outside the scope of the current manuscript. In my view it not possible to interpret the data properly without many of these additional experiments.

For example, in response to one comment, the authors have added the statement "These results suggest that the BBB opening in PAG after LEI-106 administration might be independent from endocannabinoid signaling or caused by off-target effects". I agree with the added statement, but if the authors cannot say whether the effects measured are dependent on endocannabinoid signalling or not, this significantly undermines the narrative of the manuscript.

In my original review I suggested that the manuscript was not suitable for *Physiol.J.* in its present form, and whilst I appreciate the authors have made some edits to the text to soften some claims and add some caveats, I think overall that the manuscript is not significantly improved from the original version without the suggested additional data, and still would not be suitable for *Physiol.J.* in my opinion.